# Intensifying tropical cyclones in the Arabian Sea replenish depleting aquifers
Hassan Saleh [1], Mohamed Sultan [1] ✉, Eugene Yan[2], Himanshu Save[3], Hesham Elhaddad[1,4], Hadi Karimi [1], Karem Abdelmohsen [4,5], Mustafa K. Emil[1] & Sara Al Qamshouai[6]

Tropical cyclones intensified globally in recent decades, delivering extreme precipitation deeper inland. While much research has focused on the role of climate change in tropical cyclone intensification, less is known about their contribution to groundwater recharge, especially in arid regions where freshwater is scarce and aquifers are being depleted. Here we quantify cyclone-driven groundwater recharge across the Arabian Peninsula from 2002 to 2021 using satellite-based total water storage and hydrodynamic modeling. Findings show that cyclones contributed up to 60% of total precipitation in the southern Arabian Peninsula. Cyclone Mekunu (2018) alone delivered 30 km$^3$ of precipitation inland, resulting in a net groundwater recharge of 3.2 ± 1.2 km$^3$ in the Najd subbasin. These findings reveal that tropical cyclones play a crucial role in replenishing groundwater resources in arid regions. Our approach provides a framework for quantifying recharge in ungauged arid basins worldwide, offering valuable insights for climate-resilient water resource management.

Intense tropical cyclones (TCs) have become more frequent worldwide, bearing dangerous consequences for unprepared communities[1]. In the Arabian Sea (part of the North Indian Ocean basin), a regional climate shift has increased intense TCs five-fold after 1995 compared to the previous 25 years[2]. Another study reported an increase in the frequency and duration of TCs in the Arabian Sea by 52% and 80%, respectively[3]. Despite historically experiencing fewer TCs than the other TC basins (e.g., the North Atlantic and East Pacific), the Arabian Sea witnesses the highest proportion of TCs making landfall[4]. TCs originate in the Arabian Sea, propagate towards the Arabian Peninsula (AP), and hit its southern and eastern coasts. Between 1990 and 2019, the AP encountered 12 TCs where maximum sustained wind speeds (MSW) were 65 km hour$^{-1}$ or more, some of which delivered substantial precipitation far inland[5]. For example, the extremely severe cyclonic storm (Mekunu 2018) unleashed 617 mm of torrential rain measured in the coastal city of Salalah[6]. Mekunu caused floods that impacted the dunes field in the heart of the AP and formed lakes[7] in the Rub Al Khali desert (Supplementary Fig. 1), which lasted for weeks.

TCs could potentially provide an opportunity to replenish depleting groundwater resources in the southern AP and other regions vulnerable to TCs, including Australia[8], the southwest United States[9], sub-Saharan countries[10], and western Africa[11]. We assessed the impact of TC precipitation (TCP) on the hydrology of the AP and the Najd subbasin (200,000 km$^2$) in particular (Fig. 1a) as an example of arid regions

experiencing extreme TCP. The Najd subbasin's climatic, geologic, and hydrogeologic settings are described in Supplementary Note 1 and shown in Supplementary Fig. 2.

Understanding the impact of TCP on the hydrology of the AP is essential for several reasons: (1) Intense TCs are projected to increase due to global warming[12], (2) Groundwater resources, the primary source of fresh water for irrigation and domestic use in the AP, are being depleted due to excessive extraction by its increasing population[13]; a decline in groundwater levels (GWL) (0.73 m yr$^{-1}$ between 1992 and 2010) was associated with multi-decadal desert reclamation projects in Oman (part of the Najd subbasin)[14], and a similar decline of 0.5–2 m yr$^{-1}$ was reported in Saudi Arabia[15], and (3) The impact of TCs on recharging the AP's aquifers has not been adequately investigated. Several studies provide evidence for modern recharge in the unconfined aquifers within the Najd subbasin and the confined aquifers that crop out in the mountainous areas south of the subbasin[16–18]. However, recharge from TCs and their contributions to precipitation over the AP have not been investigated. Estimating modern recharge from TCs over large areas, such as the Najd subbasin, presents considerable challenges using traditional methods (tracers, physical modeling, and numerical modeling; refer to Supplementary Note 2). To understand TCP partitioning behavior and quantify recharge with independent methods, we compared recharge estimates from the Gravity Recovery and Climate Experiment terrestrial water storage (GRACE$_{TWS}$)

[1]Department of Geological and Environmental Sciences, Western Michigan University, Kalamazoo, MI, USA. [2]Environmental Science Division, Argonne National Laboratory, Argonne, IL, USA. [3]Center for Space Research, University of Texas at Austin, Austin, TX, USA. [4]Geodynamics Department, National Research Institute of Astronomy and Geophysics (NRIAG), Helwan, Cairo, Egypt. [5]School of Sustainability, Arizona State University, Tempe, AZ, USA. [6]School of Sustainable Engineering and the Built Environment, Arizona State University, Tempe, AZ, USA. ✉e-mail: mohamed.sultan@wmich.edu

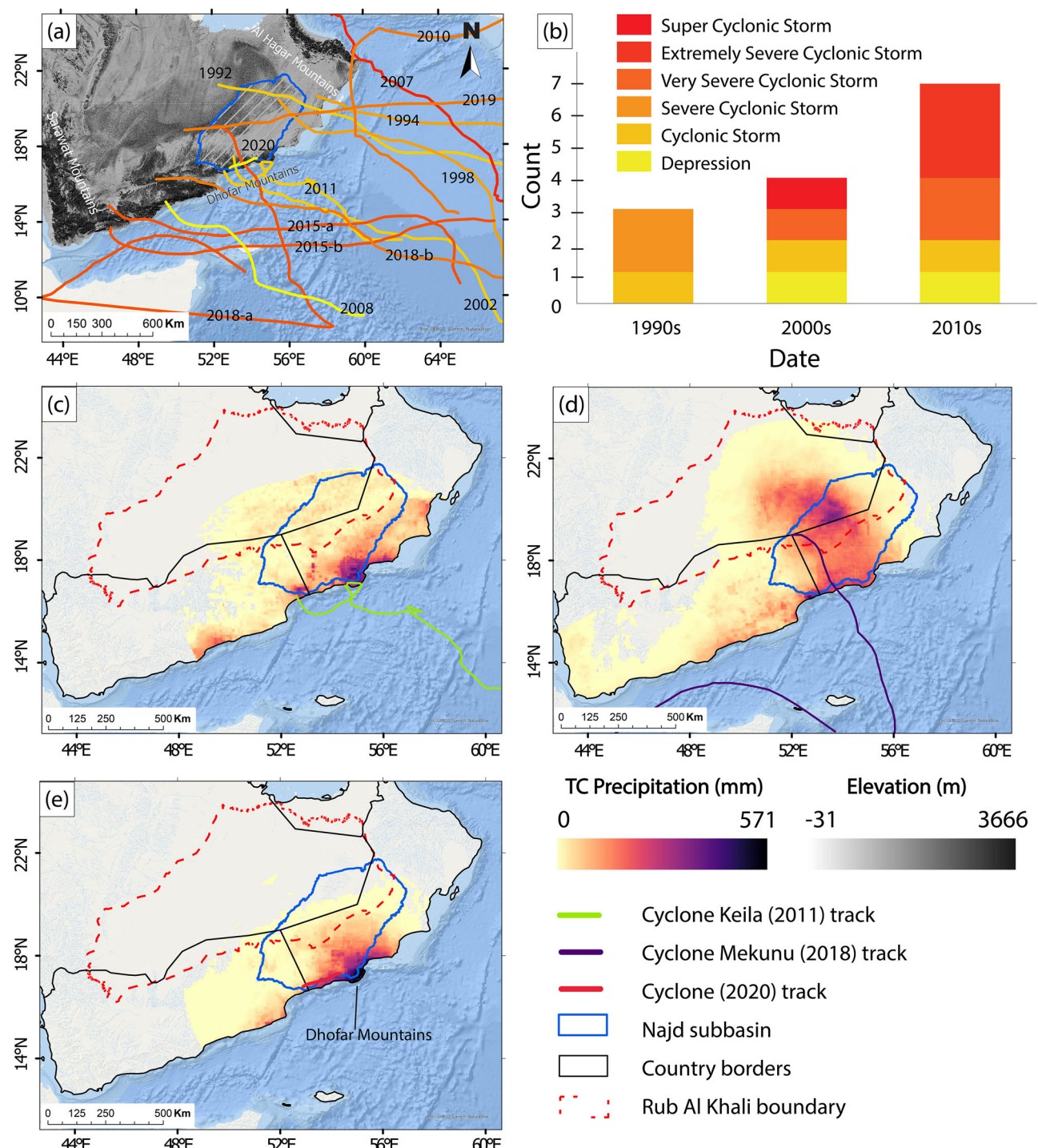

**Fig. 1 | Tropical cyclones impacting the Arabian Peninsula. a** Tracks and intensities of TCs that made landfall in southern AP between 1990 and 2020, with landfall years labeled. TC maximum intensity is color-coded from yellow (lowest intensity) to red (highest intensity). **b** Stacked bar plot of TC's decadal frequency and intensity.

**c** Cumulative precipitation from Cyclone Keila (31 Oct–8 Nov 2011). **d** Cumulative precipitation from Cyclone Mekunu (24–30 May 2018). **e** Cumulative precipitation from Cyclone 2020 (29 May–3 Jun 2020). The Rub Al Khali desert is outlined in red dashed lines, and the Najd subbasin in blue solid lines.

dataset with estimates from hydrodynamic modeling[19], coupled with the soil evaporation capacitance (SEC) model. In arid areas, stream gauges in ephemeral streams are scarce due to the sporadic nature of precipitation[20]. This limitation complicates the calibration of hydrological rainfall-runoff models for extensive TCP. We use a calibration approach based on runoff travel distances within the main ephemeral streams (Wadis).

This study quantifies TC contributions to precipitation in the AP and estimates modern recharge from three TCs that made landfall between 2002

and 2021 in the Najd subbasin, using temporal GRACE$_{TWS}$ data and hydrodynamic rainfall-runoff modeling (RiverFlow2D)[21]. The model is calibrated against satellite observations. Recharge estimates from GRACE and modeling are then validated against in situ GWL measurements. Our findings demonstrate the cruciall role TCs play in drought mitigation and groundwater recharge and provide a reliable integrated approach for assessing recharge from extensive precipitation in general and from TCs in particular in ungauged basins worldwide.

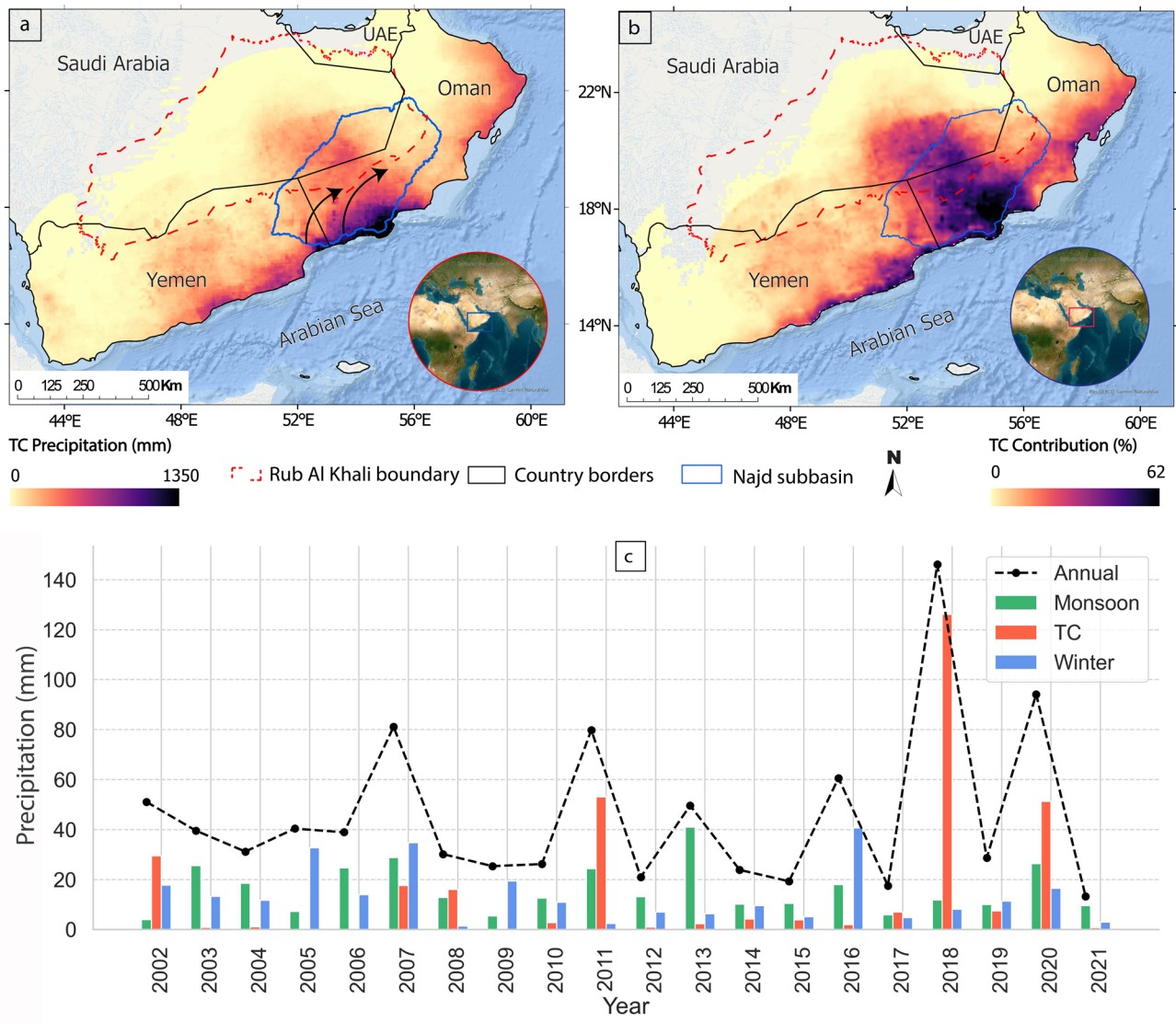

**Fig. 2 | Spatial distribution of TC precipitation over the Arabian Peninsula and its seasonal contribution over the Najd subbasin. a** Total TCP across the southern AP between April 2002 and September 2021 showing maximum accumulation over the Najd subbasin (outlined in blue), which comprises part of the Rub Al Khali desert (outlined in red dashed lines). TC intensity is color-coded from yellow (lowest accumulation) to black (highest accumulation). **b** Percentage of total precipitation attributed to TCs during the same period, with the highest relative contributions over the Najd subbasin. TC contribution is color-coded from yellow (lowest percentage) to black (highest percentage). **c** Comparison of seasonal precipitation contributions, monsoon (green bars), winter (blue bars), and TC (red bars), to the annual precipitation (black dashed line) in the Najd subbasin.. Three major cyclones impacted the Najd subbasin during the study period: Cyclone Keila (2011), Cyclone Mekunu (2018), and Cyclone (2020).

## Results

### TC characteristics and their contribution to precipitation

The dates, durations, MSW, and categories of 14 TCs that impacted the southern and eastern coasts of the AP in the past three decades (1990–2020) are summarized in Supplementary Table 1, and their tracks, which represent the centers of each cyclone, are displayed in Fig. 1a. The mountainous topography has influenced the paths of TCs that reach the southeast coast of AP. Where topographic barriers are absent, TCs moved inland regardless of their intensities, reaching distances up to 300 km. In contrast, only intense TCs propagated beyond the high mountain chains (height: up to 2600 m) along the southern coast.

Inspection of Fig. 1b and Supplementary Table 1 shows that the number of TCs that impacted the AP has progressively increased over the past three decades (1990–2000: 3; 2001–2010: 4; and 2011–2020: 7), and so did their intensities. For example, three intense TCs (MSW > 65 km h$^{-1}$) were reported between 1990 and 2000, three between 2001 and 2010, and six between 2011 and 2020. The most intense was Gonu (2009), a super cyclonic

storm that impacted the eastern coast of Oman and recharged local aquifers[22]. Three TCs brought extensive precipitation inland; Cyclone Keila (2011) made landfall twice and impacted three countries (Oman, Saudi Arabia, and Yemen; Fig. 1c). Cyclone Mekunu (2018) had the largest areal extent and introduced precipitation as far as 800 km from the coastline (Fig. 1d). In 2020, a tropical depression (Cyclone 2020), which was not named, delivered intense precipitation focused on the Dhofar mountains and triggered flooding in Najd (Fig. 1e).

The intensity of TCP from April 2002 to September 2021 over the southern AP is displayed in Fig. 2a, and its contribution to total precipitation in Fig. 2b. TCP contributions gradually decrease from the coast toward the interior, ranging from 10% to 50% along the coast, except for the southwest. The Najd subbasin is the most impacted by TCs, receiving the highest TCP intensity (1350 mm) and the highest TCP contribution ( > 60%) to total precipitation.

The cyclones that delivered substantial amounts of precipitation to the Najd subbasin were Mekunu (2018: 20.1 km³), Keila (2011: 9.7 km³), and

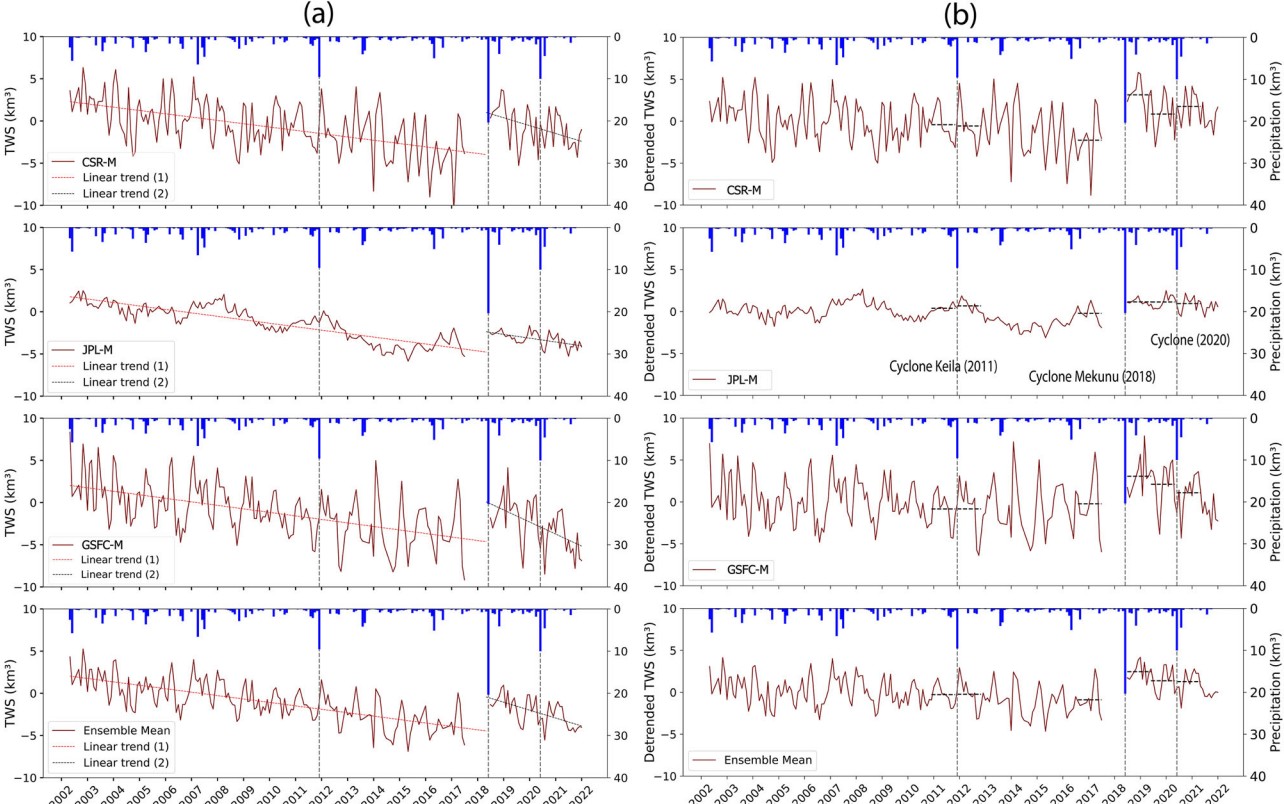

**Fig. 3 | GRACE_TWS trends and responses to TCs over the Najd subbasin. a** Time series of monthly terrestrial water storage anomalies from three GRACE solutions (CSR, JPL, and GSFC) and their ensemble mean. Monthly precipitation is shown in blue bars, with vertical dashed grey lines representing months during which TC impacted the Najd. Linear trends are shown in a red dotted line (period 1) and a black dotted line (period 2). **b** Detrended GRACE_TWS anomalies for each solution and the ensemble mean. Horizontal black dashed lines indicate the mean change in GRACE_TWS for the year preceding and following each of the three major TCs (Keila, Mekunu, and Cyclone 2020).

Cyclone 2020 (8.9 km³). A comparison of precipitation contributions from TCs, monsoon, and winter seasons (Fig. 2c) shows that the highest annual rainfall intensities occurred during cyclone years (2011, 2018, 2020). In these years, TC contributions far exceeded those from monsoons and winter precipitation, and TC contributions in other years (Fig. 2c). Mekunu delivered a 100 mm of rainfall over the Najd subbasin in three days, nearly three times its average annual precipitation. Mekunu accounted for 67% of the precipitation in 2018, Keila 61% in 2011, and 48% for Cyclone 2020. Moreover, we examined daily precipitation data and found that non-TC precipitation contributed less than 5% of the total monthly precipitation during the month of each cyclone's landfall. This indicates that the GRACE signal during these periods is dominated by TC-related precipitation.

**TC-induced recharge from GRACE and field data**

Figure 3a shows a continuous groundwater depletion before and after the landfall of Cyclone Mekunu in 2018. For example, the mean of the three GRACE mascon solutions (CSR, JPL, and CSR) is depleted at $-0.4 \pm 0.01$ km³ yr⁻¹ and $-0.9 \pm 0.4$ km³ yr⁻¹ before and after the landfall, respectively. The differences between the detrended average GRACE_TWS for the years preceding and following the three TC events (Fig. 3b) depict the changes in GRACE_TWS associated with cyclone landfall. Cyclone Mekunu had the highest impact (precipitation: 20.1 km³), marking a $3.2 \pm 1.4$ km³ increase in water storage during the study period. Cyclone Keila (2011) and Cyclone 2020 recorded precipitation of 9.7 km³ and 8.9 km³, respectively, but showed no noticeable increase in water storage (Keila: $0.1 \pm 0.2$ km³; Cyclone 2020: $-0.1 \pm 0.8$ km³) (Ensemble mean, Fig. 3b). The estimated changes in GRACE_TWS from the three mascon solutions after each of the three cyclones are provided in Supplementary Table 2.

The observed changes in GRACE_TWS from the CSR mascon solution are consistent with precipitation volumes (Fig. 4). A remarkable increase in TWS was observed following the excessive precipitation (20.1 km³) associated with cyclone Mekunu compared to the modest rise in TWS following precipitation delivered by Cyclones Keila (9.7 km³) and Cyclone 2020 (8.9 km³). The observed increases in TWS are centered over the upstream mountainous areas of the Najd subbasin, where the confined Umm Er Radhuma (UER) aquifers (Aquifers B, C, and D) crop out, covering an approximate area of 5000 km² (Supplementary Fig. 2).

Groundwater recharge, inferred from the analysis of temporal GRACE_TWS data, is corroborated by the rise in GWL in monitoring wells tapping the unconfined (Aquifer A) and the confined aquifers (Aquifers C and D) (Fig. 5). Figure 5a shows a general flow direction from the southwest, where the aquifers crop out in the high-elevation areas, towards the northeast, and Fig. 5b displays GWL variations between March 2018 (before Cyclone Mekunu) and August 2021 (after the 2020 cyclone). The water table in the unconfined aquifer (Aquifer A) rose by 1.7 meters following Cyclone Mekunu and Cyclone 2020 and by a much smaller rise (0.25 meters) following Cyclone Keila in 2011 (Fig. 5c). As most of the monitoring wells in the confined aquifers are assigned "Umm Er Radhuma" without differentiating between the distinct aquifers, we consider them as one unit in our analysis. For the confined aquifers, GWL rose gradually over two years post-Mekunu. They continued their rise following Cyclone 2020, reaching a maximum rise of 55 m in February 2024 in the Hanfit area (Fig. 5d).

A negative linear trend was found between the maximum rise in GWL following Cyclone Mekunu and the distance from the confined aquifer outcrops (Fig. 5e). The wells are clustered in two locations, Hanfit and Helat Ar Raka. The Hanfit area witnessed a maximum rise of 55 m in GWL

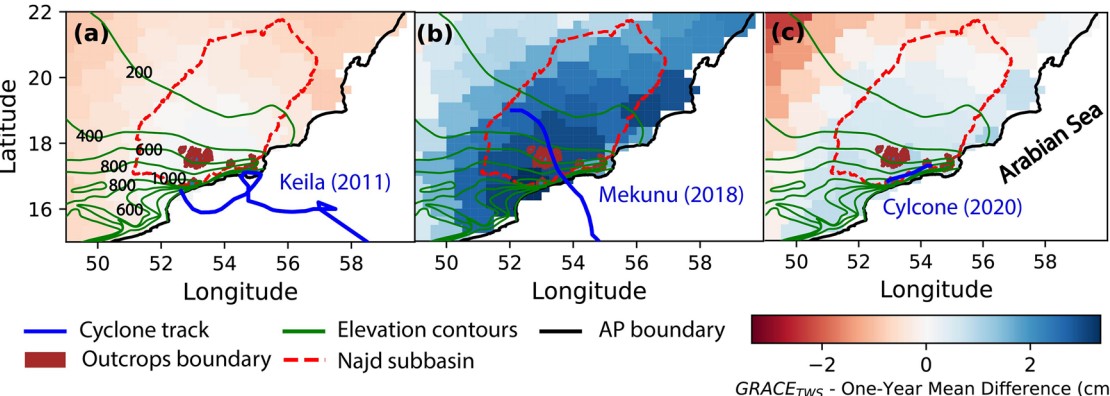

**Fig. 4 | GRACE-derived groundwater storage changes following the landfall of three major TCs.** One-year difference images of mean GRACE$_{TWS}$ (CSR solution) for (**a**) Cyclone Keila, (**b**) Cyclone Mekunu, and (**c**) Cyclone 2020. The maps depict recharge signals over the Najd subbasin and surrounding regions. Confined aquifers crop out in the upstream area. Green contour lines represent elevation, and blue lines show the tracks of the three cyclones.

following Cyclone Mekunu, while only a 3-meter rise was observed in Helat Ar Raka.

## Model calibration, validation, and evaluation
We simulated Cyclone Mekunu's TCP partitioning for 17 days using Riverflow2D and calibrated the model using satellite imagery after the cyclone's dissipation. The imagery captured the progression of runoff from high-elevation areas in the south to the northeast in two main wadis within the Najd subbasin. Calibration was achieved by adjusting the curve numbers (CNs) of the SCS Curve Number method and comparing the modeled runoff travel distances with satellite-derived distances on the dates where observations were available (Supplementary Table 3). The calibrated CNs for the simulation of Mekunu were 90 for carbonates, 75 for alluvium, and 80 for the dunes and the karstified carbonates. Model validation involved the simulation of Cyclone Keila (2011) for 18 days and Cyclone (2020) for 17 days (Supplementary Table 4) using the calibrated CNs. The calibration and validation results are summarized in Supplementary Table 5 and displayed in Fig. 6.

Statistical metrics (Fig. 6a, Inset Table) illustrate the comparison between modeled and observed runoff travel distances along the two main wadis for the three cyclones, showing consistent model performance across varying cyclone intensities (Fig. 6b). Statistical evaluation metrics demonstrated reasonable agreement between modeled and observed values with a Nash-Sutcliffe model efficiency coefficient (NSE) of 0.69, coefficient of determination (R-squared) of 0.69, and a Percent Bias (PBias) of −3.42 (Fig. 6a, inset table).

## Modeled recharge (Riverflow2D) and soil evaporation (SEC) model
RiverFlow2D results (Table 1) show the partitioning of TCP for the three cyclones that impacted the Najd subbasin. The infiltration volumes (which included both soil moisture and recharge) was 4.6 km$^3$ for Cyclone Keila, 8.4 km$^3$ for Cyclone Mekunu, and 3.4 km$^3$ for Cyclone 2020, which account for 47%, 42%, and 38% of the total precipitation, respectively. Most of this infiltration evaporated later due to the high potential evaporation in the region.

Figure 7 shows the accumulated soil evaporation estimated from the SEC model for the soil types that cover the Najd subbasin. Soil evaporation shows high evaporation rates in the first stage, which lasts for a few days, with a steep curve, followed by the second stage, with a near-flat curve. Evaporation varied by soil type, with the highest losses from carbonates following the landfall of Cyclone Keila and Cyclone 2020. For Cyclone Mekunu, carbonates and dunes contributed similarly to total evaporation losses. Soil evaporation curves are shown for Cyclone Keila (Fig. 7a), Cyclone Mekunu (Fig. 7b), and Cyclone (2020) (Fig. 7c). The simulation

continued until there was no water content available for evaporation. The total soil evaporation for Cyclone Mekunu was the highest (4.9 km$^3$), followed by Cyclone Keila (3 km$^3$), then Cyclone 2020 at 2.2 km$^3$ (Table 1). These results are consistent with TCP precipitation volumes and their temporal and spatial extent. The modeled recharge for Cyclone Mekunu (3.5 km$^3$) is similar to the measured GRACE$_{TWS}$ recharge (3.2 ± 1.4 km$^3$). The modeled and observed recharge for Keila (Model: 1.6 km$^3$; GRACE$_{TWS}$: 0.1 ± 0.2 km$^3$) and 2020 (Model: 1.2 km$^3$; GRACE$_{TWS}$: −0.1 ± 0.8 km$^3$) are low compared to Mekunu (Table 1).

## Discussion
TCs making landfall in the southern AP between 2002 and 2021 were mapped, and their contribution to precipitation was estimated. Intense TCs impacting the southern AP have become more frequent, increasing the likelihood of their propagation further inland. In contrast, weaker TCs tend to dissipate when encountering elevated terrains in the eastern and southern regions, limiting precipitation to the mountains. This orographic influence can also deflect TC tracks, as observed with Cyclone Keila in 2011. Similar behavior has been documented in numerical simulations of TCs crossing Taiwan, where orographic blocking caused deflection and created discontinuous tracks[23]. With more intense TCs impacting the southern AP, precipitation is more likely to reach arid and hyperarid areas past the mountains to the North and contribute to groundwater recharge.

The Najd subbasin received substantial contributions from TCP during the landfall of three major TCs in 2011, 2018, and 2020. In some areas, TCP accounted for more than 60% of the total precipitation, a substantial proportion given the region's low average annual rainfall ( ~ 40 mm). This contribution is tied to the rising intensity of TCs over the past three decades. As described in the section "TC characteristics and their contribution to precipitation", analysis of the MSW between 1990 and 2020 revealed that the number of intense TCs impacting the AP doubled in the last decade compared to the previous two. Our observations are consistent with those of Deshpande et al. (2021)[3], who reported a marked increase in the intensity, frequency, and duration of TCs over the Arabian Sea between 1982 and 2019.

We compared the temporal variations in GRACE$_{TWS}$ and GWL following the landfall of cyclones Mekunu and Keila to better understand recharge mechanisms and the influence of intense and moderate cyclones on groundwater in Najd. Cyclone 2020 was excluded from the comparison due to difficulties isolating its impact from Mekunu's on the observed GWL. However, the aquifer response to Cyclone 2020 is expected to resemble Keila's. The depicted depletion in GRACE$_{TWS}$ over the Najd was due to excessive groundwater extraction for agricultural development[14], where the cultivated area increased from 5 km$^2$ in 2005 to 95 km$^2$ in 2023 (Supplementary Fig. 3).

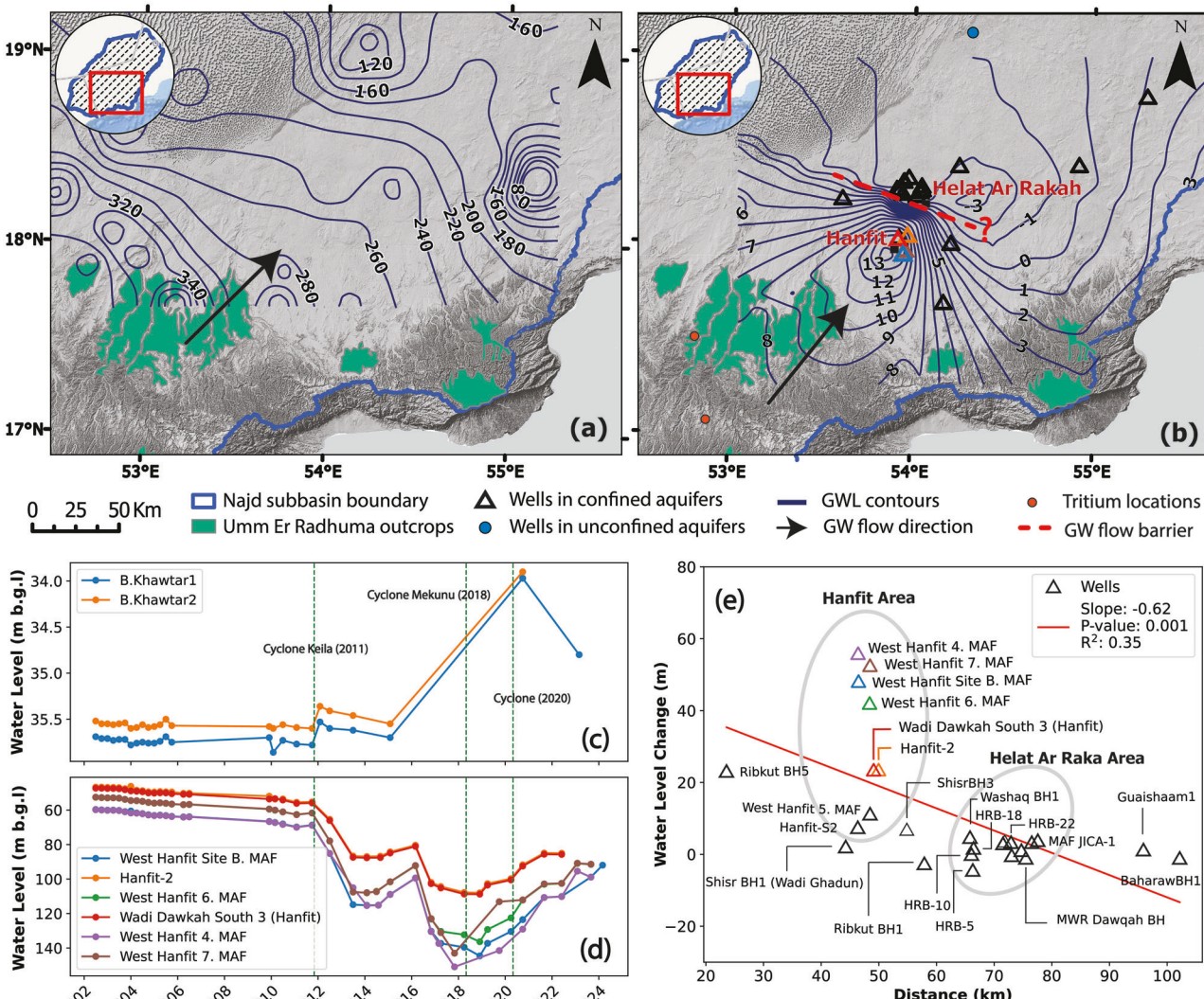

**Fig. 5 | GWL dynamics and responses to recharge in the central Najd region.**
**a** Groundwater flow directions based on pre-2011 GWL measurements from
monitoring wells and boreholes, modified from Al-Mashaikhi (2011)[16]. GWLs in
aquifer C are depicted by purple contour lines, and confined aquifer outcrops are
shown in green polygons. **b** Spatial changes in GWL between March 2018 (before
Cyclone Mekunu) and August 2021 (after Cyclone 2020). The blue solid circle marks
the location of unconfined aquifer wells, open triangles show locations of confined

aquifer wells, and orange solid circles show locations of wells where tritium was
detected. The red dashed line represents the speculated groundwater flow barrier
between Hanfit and Helat Ar Raka. **c** Time series of GWL from unconfined aquifer
wells. Vertical dashed lines represent the months when the three major TCs made
landfall. **d** Time series of GWL from confined aquifer wells. **e** Relationship between
the distance of monitoring wells from the confined aquifer outcrops and the max-
imum GWL rise following Cyclone Mekunu.

Groundwater recovery following the landfall of Cyclone Mekunu in
2018 was detected from an increase in GRACE$_{TWS}$ amounting to
$3.2 \pm 1.4$ km³. This value represents a one-year mean change in groundwater
storage over the Najd subbasin rather than short-term variations. GRACE
signals reflect both water that has percolated deeper into the ground and
reached the water table, and water still percolating slowly in the unsaturated
zone below the evaporation depth, potentially taking months or years before
reaching the water table. The postulated recharge and recovery are sup-
ported by monitoring well data. We observed a rise in GWL in the
unconfined aquifer (Aquifer A) by 1.7 m in two wells (B. Khawtar 1 and 2;
Fig. 5c) after Cyclones Mekunu and 2020. Similarly, we observed a max-
imum rise in GWL in the confined aquifer (Fig. 5d) around the Hanfit area
in several wells ranging from 6.8 to 55 m down-gradient from UER outcrops
(Fig. 5b). In contrast, Cyclone Keila showed a modest rise in GWL (0.25 m)
in the unconfined aquifer and none in the confined aquifers. The differences
in precipitation amounts and distribution of aquifer outcrops can explain
the difference in recharge magnitudes between Mekunu and Keila. Meku-
nu's precipitation was more than twice that of Keila. Also, the intense

(> 100 mm) precipitation extended over large areas (3872 km²) of the UER
aquifer outcrops (area: 5000 km²) in the case of Mekunu, compared to a
much smaller area (1973 km²) for Keila.

The spatial distribution of mass changes from GRACE after Mekunu
(Fig. 4b) shows high mass gains centered over the UER Fm outcrops in
southwest Najd, consistent with the reported recharge of confined aquifers
through UER outcrops. One interpretation of this observation is that intense
precipitation associated with cyclones over aquifer outcrops produces
aquifer recharge, leading to an increase in mass. The persistence of the
observed increase in GRACE$_{TWS}$ for multiple years following the landfall of
Mekunu supports this suggestion (Fig. 3b). If the UER aquifers are being
recharged through the outcrops in the southwest, one would expect the rise
in GWL along the groundwater flow directions to correlate with the distance
of wells from the outcrop. As shown in Fig. 5e, this relationship is evident
with a significant negative trend. By March 2019, ten months after
landfall, a notable response was observed in monitoring wells located at
the Hanfit area, approximately 50 km from the recharge zones. A sub-
stantial rise in GWL (up to 55 meters) was recorded in the confined

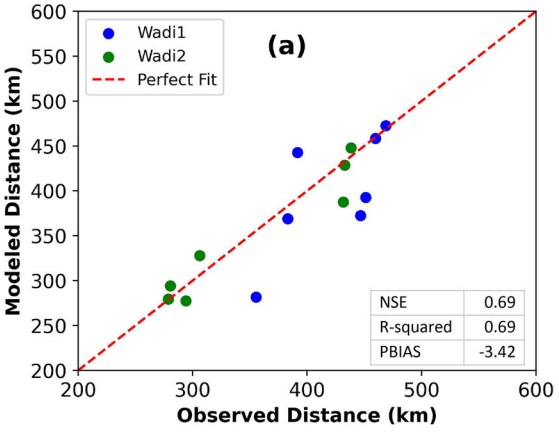
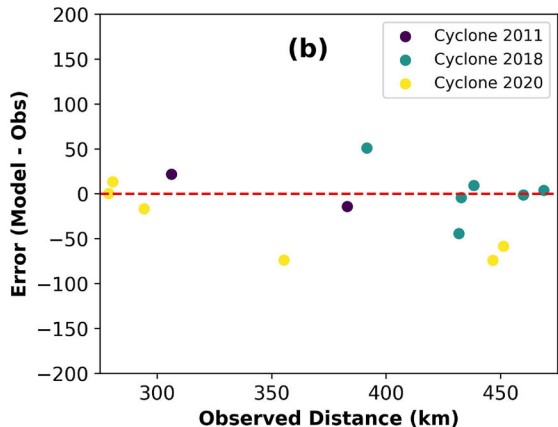

**Fig. 6 | Comparison of modeled and observed runoff travel distances for cyclones impacting the Najd subbasin. a** 1:1 scatterplot comparing modeled and observed distances for two wadis across the three simulated tropical cyclones. The inset table shows statistical evaluation metrics including NSE, R-squared, and PBIAS. **b** Residual errors between observed and modeled runoff travel distances for Cyclones Keila (2011), Mekunu (2018), and Cyclone (2020).

**Table 1 | Precipitation partitioning results (in km³) from the hydrodynamic model showing the volumes of simulated runoff, surface evaporation, and infiltration for Cyclones Keila (2011), Mekunu (2018), and Cyclone (2020)**

| TC | Rainfall | Runoff | Surface Evaporation | Infiltration | Soil Evaporation | Modeled Recharge | Observed Recharge |
|---|---|---|---|---|---|---|---|
| 2011 | 9.7 | 2.3 | 2.8 | 4.6 | 3.0 | 1.6 | 0.1 ± 0.2 |
| 2018 | 20.1 | 2.9 | 8.7 | 8.4 | 4.9 | 3.5 | 3.2 ± 1.4 |
| 2020 | 8.9 | 1.7 | 3.8 | 3.4 | 2.2 | 1.2 | −0.1 ± 0.8 |

Also shown are estimates of simulated soil evaporation, modeled groundwater recharge (calculated by subtracting soil evaporation from infiltration), and observed recharge derived from GRACE$_{TWS}$ for each cyclone

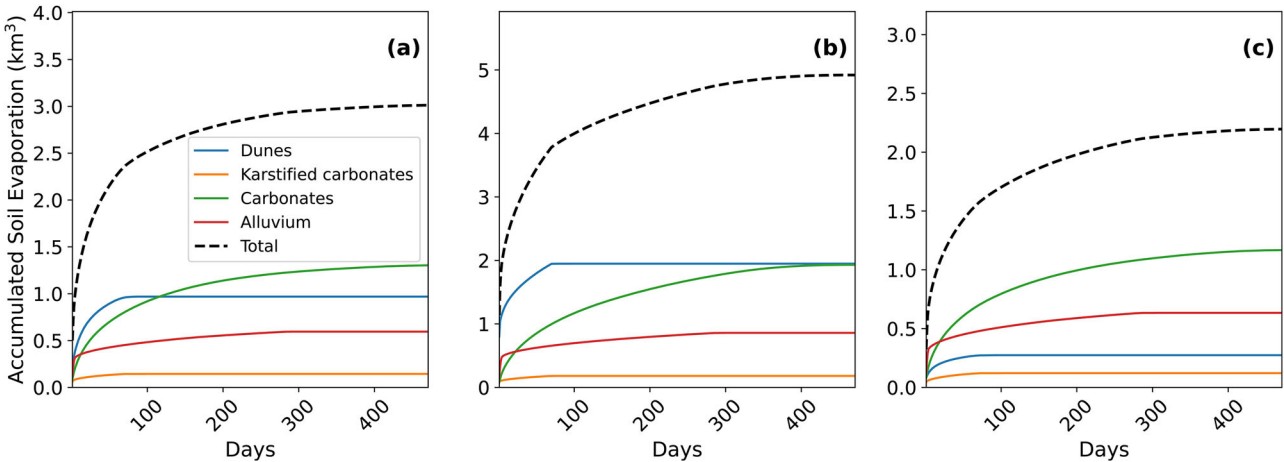

**Fig. 7 | Simulated soil evaporation by soil type following cyclone landfall. a−c** Accumulated soil evaporation (km³) simulated by the SEC model for Cyclone Keila (2011), Cyclone Mekunu (2018), and Cyclone (2020). Results are grouped by soil type within the Najd subbasin. Blue lines represent dunes, orange lines represent karstified carbonates, green lines represent carbonates, and red lines represent alluvium. The black dashed line indicates total accumulated soil evaporation.

aquifers, characterized by karst features and fractures, by February 2024. According to the Ministry of Water Resources in Oman, there was no change in pumping activity before or after Cyclone Mekunu's landfall. Therefore, we attribute the GWL rise to an increased hydraulic gradient enhanced by persistent cones of depression (≥ 50 m) in the pumping areas, which facilitates preferential groundwater flow[24]. Additionally, vertical recharge to the confined aquifers is not hydraulically possible in central Najd, where pumping activity is the highest. However, leakage to Aquifers B and C, originating from the deeper Aquifer D through faults and fractures, has been reported[16].

Further evidence for modern recharge of the confined aquifer comes from high tritium content detected in water samples collected during field campaigns in 2008 in the Najd (Fig. 5b)[16]. Tritium was detected in wells located upstream in the southwest, near the outcrop zones of the UER Fm., but not in samples collected from wells located in the northern part of the subbasin[16]. This spatial pattern indicates that modern recharge to the confined aquifers occurs through the southern outcrops and highlights the hydrogeologic control on recharge. Subsequently, groundwater flows laterally from the outcrops downgradient in a northeast direction, discharging into the Umm as Samim sabkha[25,26]. Several springs were mapped

downstream from the recharge area (Supplementary Fig. 2a) and were found to have stable isotope signatures matching deep groundwater from artesian wells in the confined aquifers[27].

One way to explain GWL rise in the central Najd is a pressure wave propagation. However, applying the pressure front equation (Supplementary Note 2), we estimated an average transient time of 24.3 days for a pressure front to reach the monitoring wells following recharge. This means that if a pressure front propagated and reached the monitoring wells, it would have occurred 9 months before the first recorded rise in GWL. Moreover, a pressure front occurs instantaneously and decays shortly, which indicates that it cannot explain the observed continuous rise over 4.5 years. These observations lead us to conclude that the measured rise was not a result of a pressure wave, but was due to actual groundwater flow. Following vertical recharge and rapid percolation through preferential flow paths to the water table, the head gradient must have increased, leading to a pressure increase and pushing groundwater in a piston-flow mechanism toward pumping-induced cones of depression in the central Najd. A similar process was described for the Lez karst system, where deep groundwater is pushed by newly infiltrated water, following intense rainfall, towards the Lez spring[28].

The unavailability of monitoring wells in recharge areas in the Najd prevents direct observation of the initial rise in GWL for estimating groundwater flow rates in the confined aquifers. The groundwater model developed by Muller et al. (2012)[25] estimated an average diffuse flow velocity of 0.5 m y$^{-1}$, but did not account for high transmissivity ( > 10,000 m$^2$ day$^{-1}$)[17] fractures and dissolution channels. Incorporating dual-permeability conditions (fractures and matrix) would explain rapid and sustained responses tens of kilometers from recharge zones. Muller also concluded that modern recharge is necessary to maintain the current GWL in central Najd, supporting our interpretation of recharge dynamics involving preferential flow[25].

The observed rapid and high rise of GWL was reported in other confined and unconfined karst aquifers. For instance, the Madison aquifer, a confined carbonate aquifer in South Dakota, saw a 34 m GWL rise during the wet period from 1990 to 1998 at the Reptile Gardens well located downstream from the recharge area[29]. Similar behavior was observed in the confined section of the Edwards confined karst aquifer in Texas, where GWL rose by 31 m over two years[30]. In the Ombla catchment in Croatia, a GWL rise ranging between 52 and 128 m over 3.5 years was observed[31]. Such cases demonstrate that rapid responses in karst carbonate aquifers are possible under structural and hydrogeological controls. In Najd, notable GWL rises were limited to the Hanfit area and not Helat Ar Raka further downgradient (Fig. 5a, b), likely due to a structural or lithologic barrier impeding flow and causing accumulation of groundwater upgradient. This suggestion is supported by the observed steep gradient of the contour values between the two areas (Fig. 5b).

In our conceptual model, groundwater recharge from TCP occurs predominantly through focused mechanisms associated with geological structures and karst features. While shallow soil ( < 1 m) and alluvium ( ~ 4 m) contribute minimally to overall aquifer storage, recharge into the unconfined to semi-confined (Aquifer A) occurs locally where wadis incise the aquifer[32]. In contrast, the UER confined aquifers are primarily recharged at their southern mountain outcrops, where intense TCP intersects highly karstified and fractured carbonate formations. Following recharge, a pressure wave may instantaneously propagate in the aquifer, increase the GWL in the wells, then decay quickly[33] before any measurements are taken in the monitoring wells. Groundwater likely flows in a piston-flow mechanism and discharges at Umm as Samim sabkha and dispersed springs northeast of the Najd. While diffuse recharge is very slow (0.5 m y$^{-1}$)[25], preferential groundwater flow is rapid along highly solution-channeled fractures and fault systems[17]. GWL in wells located approximately 50 km from recharge areas exhibit delayed responses of at least 6–10 months following cyclone landfall (Fig. 5d). Vertical leakage from Aquifers A and B downward into deeper Aquifers C and D is not possible in the central Najd due to higher hydraulic head pressures in these deeper confined aquifers[24]. Conversely,

upward leakage from Aquifer D into the overlying Aquifers and groundwater mixing along faults are possible[16], further emphasizing the role of lateral preferential flow and upward leakage as the main recharge mechanisms.

The calibrated model closely replicated the runoff observed from satellite imagery. We noticed variations in the distances traveled by channeled water along the two wadis during the activity of the three TCs. More intense TCP resulted in longer travel distances, while weaker TCP produced shorter distances. Statistical metrics, including an NSE of 0.69 for runoff distances across the three cyclones (Fig. 7), confirm the model's effectiveness in simulating TCP partitioning[34]. The modeled infiltration comprises two components: soil moisture and groundwater recharge. Soil evaporation, a major component of the water cycle in arid regions, accounted for an average of 63% of the infiltrated water for the three simulated TCs. After accounting for soil evaporation, the modeled recharge for Cyclone Mekunu was in good agreement with the recharge estimate from GRACE$_{TWS}$. However, the modeled recharge for Cyclone Keila and Cyclone (2020) exceeded the values observed by GRACE$_{TWS}$. We attribute this discrepancy to the relatively low recharge volumes during these two events, resulting in minimal mass changes, which likely fell below the measurement threshold by GRACE. Previous studies report that the smallest resolvable basin size for GRACE is approximately 63,000 km$^2$ with an uncertainty of 2 cm in equivalent water height, which decreases when catchment size increases[35]. The Najd subbasin ( ~ 200,000 km$^2$) is more than three times larger than this threshold, making it suitable for GRACE-based groundwater recharge estimation from TCs that produce sufficient mass change. For basins of this size, the minimum resolvable TWS anomaly is 1–1.5 cm, equivalent to ~2–3 km$^3$ [35,36]. Cyclone Mekunu produced a recharge volume that exceeded this threshold, while Keila and Cyclone 2020 likely did not, which explains the difference between modeled and GRACE-derived recharge for the smaller events. As a result, recharge from lower-magnitude events is more reliably captured by our hydrodynamic model than by GRACE.

While GRACE data provide invaluable insights into regional groundwater changes, its coarse spatial (300–400 km) and temporal (monthly) resolutions limit the ability to capture localized and short-term variations, particularly during extreme events like TCs. Small-scale changes in water storage are often masked by noise, adding uncertainty to recharge estimates. Groundwater monitoring in the Najd is further constrained by sparse and manually collected well data, which lacks temporal continuity due to months-long gaps between measurements. This under-sampling introduces uncertainties in capturing the immediate contribution of TCs to groundwater recharge. Moreover, relying on manual calibration for the hydrodynamic model presents another limitation. While necessary, manual calibration may introduce biases and reduce the model's predictive reliability. Moreover, uncertainty arises from the SEC model used for estimating evaporation losses. The SEC model does not explicitly represent focused infiltration beneath wadis or fractures, but assumes uniform soil depths and properties for each soil type across the Najd. These assumptions complicate accurate evaporation modeling and ultimately affect modeled groundwater recharge volumes. Together, these factors contribute to uncertainties in the recharge estimations and emphasize the need for improved observational data to validate modeling approaches in arid regions.

## Methods

The adopted methodology involves four critical steps. Initially, we analyzed TC data over the Arabian Sea from the International Best Track Archive for Climate Stewardship (IBTrACS, version 4) database covering 1990 to 2020 to identify the TCs that made landfall, their characteristics, and their temporal and spatial variations over the AP (Step I). The following steps (II through IV) were conducted throughout the period (April 2002–September 2021), where additional data (satellite-based precipitation and GRACE/GRACE-FO) necessary for performing our analysis are available. The second step involved the selection of the subbasin that received the bulk of TCP (Step II). Then, we calculated water mass changes (groundwater recharge) over the selected subbasin following each identified TC landfall using

GRACE$_{TWS}$ data from three mascon solutions and validated the results against GWL data (Step III). Finally, we constructed a hydrodynamic rainfall-runoff model coupled with the SEC model to quantify groundwater recharge in the selected subbasin following the identified TCs, and compared the results with changes in GRACE$_{TWS}$ (Step IV).

## TC characteristics and their temporal and spatial variations

TC records were obtained from the IBTrACS database[37]. This database combines best-track data from different agencies and is commonly used for global[38,39] and regional studies in the Arabian Sea[40–42]. Several TC characteristics (i.e., location, time, wind speed, and name) were extracted from the IBTrACS records for the Arabian Sea. TC intensity, defined by maximum sustained wind speed (MSW), has been recorded since 1990. Landfall is marked when TC's centers intersect land; however, its impact on land can be observed even when the center remains 110 km offshore[37]. To analyze the frequency and spatial distribution of TCs making landfall in the AP, the individual TCs were classified based on their reported MSW using the World Meteorological Organization (WMO) regional system's classification.

## Selection of the subbasin receiving the bulk of TCP

The subbasin selected for further analysis received the highest contribution of TCP to its total precipitation. TC contribution to precipitation was estimated from all TCs that impacted the AP during the study period. The Integrated Multi-Satellite Retrievals for GPM (IMERG) is a precipitation product that utilizes satellite-based observations and available gauges to produce the gauge-corrected IMERG-Final product. We used IMERG-Final (v06), covering April 2002 to September 2021 with a spatial resolution of 0.1° x 0.1° at a daily time scale[43]. We applied the following scheme to daily gridded precipitation data during TC activity to estimate the contribution of individual TCs to the total precipitation: (1) the area of daily precipitation was defined based on a commonly applied[8,44] 500 km radius[45] around 3-hourly (interpolated to daily) track observations of TCs; (2) the duration of cyclone activity was extended three days beyond the best-track activity dates to account for precipitation from post-dissipation clouds[46]; and (3) precipitation from all TCs was aggregated as a ratio of total precipitation to represent their contribution over the study period. The Copernicus global digital elevation model (COP30m DEM)[47] was used to delineate the subbasin receiving the highest TCP contribution. Furthermore, monthly, seasonal, and annual precipitation were aggregated from the daily IMERG-Final data and averaged over the selected subbasin to identify TCs with the highest impact on the subbasin during the study period.

## TC-induced groundwater recharge from GRACE, field data, and modeling

GRACE (2002–2017) and GRACE-FO (2018–present) are two satellite missions that provided monthly measurements of water mass change spatially and temporally from Apr 2002 to Sep 2021, with a one-year gap between May 2017 and May 2018. We utilized three GRACE mascon solutions to estimate the monthly change in water storage from TC landfall in the Najd subbasin: (1) CSR-M (RL0602) mascon solution (grid size: 0.25° x 0.25°) provided by the Center for Space Research[48,49] (2) JPL-M (RL06.1) mascon solution (grid size: 0.5° x 0.5°) produced by the Jet Propulsion Laboratory[50] after applying gain factors to the gridded solution, and (3) GSFC-M (RL06v2.0) mascon solution (grid size: 0.5° x 0.5°) provided by the Goddard Space Flight Center[51]. Missing months were filled using linear interpolation[52]. Long-term linear trends (2002–2021) were calculated and removed from each time series to reveal the changes in water storage resulting from cyclone landfall[53,54]. The increase in TWS following the landfall of individual TCs was estimated by calculating the difference between the one-year mean of detrended GRACE$_{TWS}$ over the subbasin before and after each TC landfall.

The change in GRACE$_{TWS}$ ($\Delta$TWS) is composed of changes in soil moisture ($\Delta$SMS), surface water/reservoir storage ($\Delta$SWS), and groundwater ($\Delta$GWS) (Eq. 1; Rodell & Famiglietti, 2002[54]).

$$\Delta TWS = \Delta SMS + \Delta SWS + \Delta GWS \tag{1}$$

The Najd subbasin is in a hyper-arid area with no perennial rivers, lakes, or reservoirs. Most of the soil moisture associated with sparse precipitation events evaporates within days[55,56], given the area's high temperatures (up to 50 °C)[24]. The difference between the one-year mean of GRACE$_{TWS}$ over the subbasin before and after each TC landfall reflects the water that reached the groundwater table and water percolating in the unsaturated zone below the evaporation depth, hereafter referred to as recharge.

The $\Delta$TWS resulting from a TC that occurred during the one-year data gap was estimated from the difference between the one-year mean before and after the gap. We report the mean values of the three solutions (CSR-M, JPL-M, GSFC-M) to suppress the noise and estimate the relative uncertainty of the reported GRACE$_{TWS}$ measurements from their standard deviation. The estimated recharge from $\Delta$GRACE$_{TWS}$, assumed to represent $\Delta$GWS for the individual TCs, were then compared with the modeled (from the hydrodynamic model) recharge to evaluate the model results.

The temporal variations (2002–2024) in GWL provided by the Ministry of Regional Municipalities and Water Resources (MRMWR) in Oman for confined (27 wells; Supplementary Fig. 4) and unconfined (2 wells) aquifers were examined. GWL were measured at temporal intervals ranging from three to six months, with occasional longer gaps filled using linear interpolation. The distribution of GWL at any specified time was then extracted by spatial interpolation using the Kriging method in ArcGIS Pro (v3.3). Two GWL maps were created: one for March 2018, before the landfall of Mekunu, and the other for August 2021, after the landfall of Cyclone 2020. The change in GWL following TC landfall was obtained by subtracting the former from the latter. The recharge or absence of it was inferred from the analysis of temporal GRACE$_{TWS}$ data and validated by the observed variations in GWL.

## Hydrodynamic model setup

The RiverFlow2D hydrodynamic model simulated the effects of TCP in the Najd subbasin, producing fully distributed runoff, infiltration, and surface evaporation layers. The model routes runoff in wadis and simulates flood-plain inundation using shallow-water equations[21]. The model resolves complex topography with adaptive triangular-cell meshes. The mesh was constructed with variable spatial resolutions by buffering stream orders from levels 1 to 8 with a buffer radius of 100 to 800 meters, respectively. Higher-resolution meshes were assigned to high-order stream buffers, where land surface runoff reflects the main water movement. In contrast, lower-resolution meshes were used for low-order stream buffers, where stream channels predominantly route the surface water. The mesh cell size ranges between 30 m and 1000 m, with a 1200 m background resolution, providing optimized performance within computational limits. Infiltration in the model is treated as a loss determined by gravity and capillary action. However, the model does not account for soil evaporation. We estimated soil evaporation using the SEC model, where the resulting values were subtracted from the modeled infiltration to calculate recharge.

Model inputs include daily gridded precipitation data (IMERG-Final), a soil type map, and a 30 m resolution digital elevation model (COP30m) (Supplementary Fig. 5). Monthly pan evaporation rates for high exposure settings in mountain foothills and interior planes[57] were multiplied by a factor (0.75) to account for overheating by the metal sides of the pan[58] and applied across the subbasin. Infiltration was estimated using the SCS-CN method[59], with an initial abstraction ratio of 0.05[60] and antecedent moisture content (AMC1) representing dry conditions in arid areas. We assigned CN values reported in USDA-SCS (1985)[59] to soil types (alluvium, dunes, carbonates, karstified carbonates) extracted from geologic maps of the Najd (Supplementary Fig. 6a). In assigning the CNs, we were guided by ranges reported for similar soil, rock units, and settings[61]. A Manning's roughness coefficient of 0.035 was used for mountain streams with no vegetation[62] and was assumed constant for the Najd. Each simulation began when TCP

entered the Najd subbasin and ended when runoff reached the farthest travel distance in the two main wadis measured from the mountain divide. The outputs included spatially distributed maps of runoff, infiltration, and surface evaporation.

### Hydrodynamic model calibration, validation, and evaluation

Multispectral satellite sensors onboard Landsat-7 and the Harmonized Landsat Sentinel (HLS) product provide surface reflectance data with 30 m spatial resolution and varying temporal resolutions[63]. The Normalized Difference Water Index (NDWI; McFeeters, 1996[64]) was employed to delineate runoff following TC landfall. The NDWI utilizes the green and near-infrared (NIR) bands (Eq. 2) and separates water from land, with values > 0 indicating water bodies.

$$NDWI = (Green - NIR)/(Green + NIR) \qquad (2)$$

Analysis of stream networks and satellite imagery revealed that runoff primarily flows through two major wadis extending over 400 km from the mountain divide toward the lowland dunes (Supplementary Fig. 6b). The model was set for Cyclone Mekunu (2018), the TC that generated the highest precipitation volume in the Najd subbasin, and simulated its landfall. The model was calibrated by iteratively adjusting the CNs until the modeled runoff traveled distances similar to those observed from the NDWI images in the two wadis. The calibrated parameters were then applied to simulate Cyclone Keila (2011) and Cyclone (2020) to validate the model performance at varying TC intensities. Three statistical metrics, the Nash–Sutcliffe model efficiency coefficient (NSE), the Percent Bias (PBias), and the coefficient of determination (R-squared), were used to evaluate the model performance by comparing modeled and observed runoff travel distances in the two wadis for the three TCs[34].

The SEC model (Supplementary Note 3) was used to estimate soil evaporation for soil types that cover the Najd (Supplementary Tables 6, 7). The model simulates soil evaporation and leakage beyond the active evaporation depth based on potential evaporation, soil moisture content, and hydraulic properties of each soil type[65,66]. Inputs of the SEC model include (1) the fully distributed (cell-based) infiltration layer simulated by the RiverFlow2D for each TC event, (2) the hydraulic properties of soil types estimated by Lehmann et al. (2018)[55] and reported by Merlin et al. (2016)[66], and (3) the potential evaporation applied in the RiverFlow2D simulation. We assumed the maximum storage of a cell to be the product of the characteristic length of each soil type and the saturated moisture content. If infiltration exceeded the maximum storage, the cell was assigned its maximum moisture content capacity. The soil types used in the model were extracted from the Harmonized World Soil Database (v2.1)[67]. Three soil types (Loamy Sand, Sandy Loam, and Loam) were selected as equivalent to the Najd subbasin's lithologies. The SEC algorithm updates (daily) the amount of water remaining as soil moisture content based on two evaporation stages. The governing equations for stage (1) and stage (2) are described in detail in Lehmann et al. (2018)[55] and are given in Supplementary Note 3. The soil evaporation rate is often high in stage (1) and lasts for a few days, followed by a low rate in stage (2), which could last for months[55]. The SEC model simulation time was extended until the moisture content was below the residual water content value when no moisture was available for evaporation. The accumulated soil evaporation was subtracted from the modeled infiltration to estimate recharge, which was then compared to the observed recharge estimated from $GRACE_{TWS}$.

### Reporting summary
Further information on research design is available in the Nature Portfolio Reporting Summary linked to this article.

### Data availability
Remote sensing observations are publicly available. CSR (RL06.2) mascon solutions (grid size: 0.25° × 0.25°) were downloaded from https://www2.csr. utexas.edu/grace/RL0602_mascons.html/, JPL (RL06.1) mascon solutions (grid size: 0.5° × 0.5°) from https://search.earthdata.nasa.gov/, and GSFC (RL06v2.0) mascon solutions (grid size: 0.5° × 0.5°) from https://earth.gsfc. nasa.gov/geo/data/grace-mascons/. The Copernicus global digital elevation model (COP30m DEM) was downloaded from https://opentopography. org/. The IBTrACS v04 data were downloaded from https://www.ncei.noaa. gov/products/international-best-track-archive. The HLS NDWI images were processed using the Sentinel Hub API https://github.com/sentinel-hub/sentinelhub-py. GWL data were requested from the Ministry of Regional Municipalities and Water Resources (MRMWR) in Oman and are restricted to usage in this research, as per agreement. Other produced datasets are provided in the supplementary materials and deposited in fig-share at https://doi.org/10.6084/m9.figshare.28079441.

### Code availability
RiverFlow2D software (v8.12) https://www.hydronia.com/riverflow2d was used for hydrodynamic modeling. Statistical analyses were performed and figures were generated using Python (3.11) and ArcGIS Pro (v3.3). The code and files used for simulating soil evaporation are available at https://doi.org/10.5281/zenodo.14957231 (version v1.0).

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

## Acknowledgements
This project was funded by the National Aeronautics and Space Administration (NASA) Earth Science Division grant (80NSSC24K1155) awarded to Western Michigan University, and supported by the ESA Network of Resources Initiative. The first author was supported by the GSA Graduate Student Research Grant (2023). The authors thank the Ministry of Regional Municipality and Water Resources in Oman for providing the field data. We are grateful for the valuable discussions and suggestions with Khalid Al-Mashaikhi and Abdullah Ibrahim throughout this study.

## Author contributions
Hassan Saleh**:** responsible for conceptualization, formal analysis, investigation, methodology, software, validation, visualization, and writing – original draft, review, and editing. Mohamed Sultan: contributed to supervision, conceptualization, formal analysis, investigation, methodology, validation, writing, review, and editing. Eugene Yan: supervised hydrodynamic model construction, validation, and interpretation. Himanshu Save: contributed to the formal analysis and interpretation of GRACE data. Hesham Elhaddad: contributed to model construction and validation. Hadi Karimi: participated in formal analysis and interpretation. Karem Abdelmohsen: participated in formal analysis and interpretation. Mustafa Emil: contributed to satellite data processing and formal analysis. Sara Al Qamshouai: contributed to in situ data analysis.

## Competing interests
The authors declare no competing interests.
