## [Transparent Peer Review file · Communications Earth & Environment]

Intensifying tropical cyclones in the Arabian Sea replenish depleting aquifers

Corresponding Author: Professor Mohamed Sultan

Version 0:

Decision Letter:

Dear Professor Sultan,

Your manuscript titled "Intensifying tropical cyclones in the Arabian Sea replenish depleting aquifers" has now been seen by 3 reviewers, whose comments are appended below. You will see that they find your work of some potential interest. However, they have raised quite substantial concerns that must be addressed. In light of these comments, we cannot accept the manuscript for publication, but would be interested in considering a revised version that fully addresses these serious concerns. Specifically, a revised manuscript must:

1. Fully clarify the model for estimating groundwater recharge from extreme rainfall, emphasizing uncertainty in measurements.
2. Fully explain how the recharge process affects the confined aquifers.
3. Compellingly demonstrate the robustness of your methodology and consider improving data resolution, better matching cyclone tracks with the assessed area, and accounting for additional factors such as aquifer movement, pre-cyclone rainfall, and continuous groundwater monitoring.

We hope you will find the reviewers' comments useful as you decide how to proceed. Should additional work allow you to address these criticisms, we would be happy to look at a substantially revised manuscript. If you choose to take up this option, please either highlight all changes in the manuscript text file, or provide a list of the changes to the manuscript with your responses to the reviewers.

When resubmitting, please provide a point-by-point response to the reviewers' comments. Please submit your responses as a separate file, distinct from your cover letter where you can add responses to the Editors' comments that you do not want to be made available to the reviewers. Word files are preferred. We recommend that any figures, tables or graphs that are included in the response to reviewers are also included in the main article or Supplementary Information.

If the revision process takes significantly longer than three months, we will be happy to reconsider your paper at a later date, as long as nothing similar has been accepted for publication at Communications Earth & Environment or published elsewhere in the meantime.

Please use the following link to submit your revised manuscript, point-by-point response to the reviewers' comments with a list of your changes to the manuscript text (which should be in a separate document to any cover letter), a tracked-changes version of the manuscript (as a PDF file) and any completed checklist:

Link Redacted

Please do not hesitate to contact us if you have any questions or would like to discuss the required revisions further. Thank you for the opportunity to review your work.

Best regards,

Jose Luis Iriarte Machuca, PhD
Editorial Board Member
Communications Earth & Environment

Alireza Bahadori, PhD
Associate Editor
Communications Earth & Environment
Consulting Editor
Communications Sustainability

EDITORIAL POLICIES AND FORMAT

If you decide to resubmit your paper, please ensure that your manuscript complies with our editorial policies and complete and upload the checklist below as a Related Manuscript file type with the revised article:

Editorial Policy Policy requirements
(Download the link to your computer as a PDF.)

- Behavioural and social science
- Ecological, evolutionary & environmental sciences
- Life sciences

<https://www.nature.com/documents/nr-reporting-summary.zip>

For your information, you can find some guidance regarding format requirements summarized on the following checklist: (<https://www.nature.com/documents/commsj-phys-style-formatting-checklist-article.pdf>) and formatting guide (<https://www.nature.com/documents/commsj-phys-style-formatting-guide-accept.pdf>).

REVIEWER COMMENTS:

Reviewer #1 (Remarks to the Author):

The article, "Intensifying tropical cyclones in the Arabian Sea replenish depleting aquifers", by Hassan Saleh and co-authors is topical, given widespread concern around the consequences of the amplification of precipitation extremes (e.g. Fischer and Knutti, 2016) and 'global-scale' groundwater depletion observed most persistently in drylands (e.g. Jasechko et al., 2024). The manuscript describes well a thoughtful analysis showing the contributions of extreme precipitation events - tropical cyclones – to episodic groundwater replenishment in a large, arid and ungauged basin (Najd sub-basin). The authors draw upon a range of techniques and data - albeit dominated by remote-sensing - to convincingly describe this (episodic) connection. It is pleasing to see the argument further rooted in ground-based observations (piezometry) presented in Figures 5 and S3.

My main comment – and it is minor – is that the authors incorporate in a revised manuscript a more explicit discussion of (1) their conceptual model – based on the wide range of techniques they have employed - of how extreme rainfall is conveyed to unconfined and confined aquifers (e.g. focused versus diffuse recharge); and (2) its consequences for the uncertainty in the (volumetric) measurement of recharge. On this second point, Table 1 documents well (appropriately) uncertainty associated with the computation of recharge derived from TCP. In section 4.4, there are also thoughtful reflections on the challenges and uncertainties in estimating fluxes and storage changes in the Najd sub-basin. The paper would be importantly improved if it included a greater synthesis of the evidence outlining (essentially) the 'coarseness' of the analysis - explaining why they consider groundwater recharge can be estimated (quantified) for very substantial water surpluses from TCP (e.g. Mekunu, 2018) but evidently not estimated robustly from less voluminous TCP events. Is this strictly a signal-to-noise ratio in GRACE Δ TWS? If so, how does that square with their claim – see minor comment #1 below – to provide a robust framework for quantifying recharge in ungauged arid basins worldwide?

Minor comments:

1. In an otherwise carefully worded manuscript, the claim on lines 20-21 in the Abstract – “Our approach provides a robust framework for quantifying recharge in ungauged arid basins worldwide...” is – to my mind - overstated. The authors have employed a range of methods (e.g. GRACE Δ TWS, piezometry, numerical modelling) to address observational constraints (i.e. ungauged catchments) in an arid basin that is of scale aligned to the use of GRACE Δ TWS without incurring the considerable uncertainty of disaggregating and downscaling GRACE Δ TWS.
2. The authors describe on lines 31 to 33 to “... the extremely severe cyclonic storm (Mekunu, 2018) unleashed 617 mm of torrential rain measured in the coastal city of Salalah. Mekunu caused floods that impacted the dunes field in the heart of the AP and formed lakes in the Rub Al-Khali desert (Supplementary Figure 1), which lasted for weeks.” I could find no reference to the appearance of lakes in Supplementary Figure 1.
3. The authors assert on lines 34 to 35 that, “TCs could potentially provide an opportunity to replenish depleting groundwater resources in the southern AP and other regions vulnerable to TCs including Australia, the southwest United States, and the north African regions.” This also occurs in southern Africa – tropical cyclone Eline in February 2000 as reported by Cuthbert et al. (2019) in Namibia.
4. On line 200, the authors estimate (infer) “groundwater flow rates”. I suggest that the authors more precisely refer to these as “average linear velocities” to differentiate them from Darcy fluxes.

Editorial suggestions:

- Line 107: The authors refer to “UER” aquifers without explicitly defining what the acronym “UER” is.
- Line 125: Similarly, the authors refer to “CNs” without defining this acronym.
- Line 135: Similarly, the authors refer to “SEC” model without explicitly defining what the acronym “SEC” is. This occurs later on line 351.
- Line 177: Replace “were” with “are”?
- Line 200: Replace “meters” with “m”?
- Line 310: The authors refer to the hydrodynamic model, RiverFlow2D, without providing a reference or source (e.g. hydronia.com?) for this model.

Cited references:

- Cuthbert, M.O., R.G. Taylor, G. Favreau, M.C. Todd, M. Shamsudduha, K.G. Villholth, A.M. MacDonald, B.R. Scanlon, D.O.V. Kotchoni, J.-M. Vouillamoz, F.M. A. Lawson, P.A. Adjomayi, J. Kashaigili, D. Seddon, J.P.R. Sorensen, G.Y. Ebrahim, M. Owor, P.M. Nyenje, Y. Nazoumou, I. Goni, B.I. Ousmane, T. Sibanda, M.J. Ascott, D.M.J. Macdonald, W. Agyekum, Y. Koussoubé, H. Wanke, H. Kim, Y. Wada, M.-H. Lo, T. Oki, N. Kukuric, 2019. Observed controls on resilience of groundwater to climate variability in sub-Saharan Africa. *Nature*, Vol. 572, 230–234.
- Fischer, E.M. and Knutti, R. (2016) Observed heavy precipitation increase confirms theory and early models. *Nature Climate Change* Vol. 6, 986–991.
- Jasechko, S., Seybold, H., Perrone, D., Fan, Y., Shamsudduha, M., Taylor, R.G., Fallatah, O. and Kirchner, J.W. (2024) Rapid groundwater decline and some cases of recovery in aquifers globally. *Nature*, Vol. 625, 715-721.

Reviewer #2 (Remarks to the Author):

This paper is a great contribution. It has been known that these cyclones are probably contributing to recharge in these arid regions, but it's been difficult to quantify. Using GRACE data and the hydrology modeling were good ways to try to get at these numbers. I have some comments below, especially about interpreting how the recharge is affecting the confined aquifers (from which most of the pumping is occurring). I think this paper should be accepted for publication, but my comments below should be addressed.

Line 33. As written it made me look to the figure for where this lake was in the Rub Al Khali, only to be disappointed not to find it.

Figure 5 is not clear which wells are unconfined vs have tritium detected as the symbol is nearly identical.

Line 197. Although Tritium may have been detected, change in head in confined aquifers are due to a pressure wave, not the displacement of water. I can't imagine all the water in the aquifers being displaced by the rain from one cyclone. It must have been many events since the 1960s. So tritium is not evidence that this one cyclone recharged the confined aquifer outcrops substantially.

Lines 200 through 205. Such high flow rates in karst aquifers are for unconfined aquifers with spring outlets. The UER aquifers are confined with no outlets except pumping wells today. Flow rates would be orders of magnitude smaller. I worked with Thomas Muller on his PhD thesis (you cite him in SI) in this area. His groundwater model would have produced velocity estimates that could be used here. These were used to estimate travel times compared with his environmental tracer data. As a rough estimate the groundwaters move 500 km over 50k years. This translates to 10 meters per year. It may be that recent pumping is pulling the modern-water front in UER inland faster, or that there is some leakage from above that contributes enough tritium to be detected. It would be helpful to show some pumping data over time to demonstrate that the head changes in the confined aquifers are

not due to changes in pumping rates. For example, did the heavy rains lead to less pumping for some time? It must be true that the mass increase has occurred nearly all at the water table and above in the unsaturated zone. So using specific yield values for the unconfined aquifers (and storativity values for the confined aquifers) in conjunction with water level changes might also produce some crude estimate of change in storage for comparisons. It seems that the areal extent of the GRACE anomaly is mostly over the A and B aquifers rather than the outcrops of the C and D aquifers. So in spite of the peak of the anomaly being over the C and D outcrops, the majority of the mass and recharge must have been to the B and especially the A aquifer. Which leads to the question--will increasing heads in the surficial aquifer propagated downward to the unconfined aquifers over time, or are they too hydraulically disconnected to make a difference? If you can estimate the transmissivity and storativity of the UER aquifers, could you not calculate the propagation wave of an instant recharge event at the outcrop as it extends toward and reaches the wells inland?

Reviewer #3 (Remarks to the Author):

The impacts of climate change on evolution of land surface is always a widely- seeking topic nowadays. Most results show the negative results induced by climate change. This manuscript describes the intensification of Tropical cyclones (TCs) on the water resources in arid area, especially for the aquifers, and the results of more recharge of groundwater is quite clear and helpful for the water management in arid region.

However, the study has several limitations in its methodology. I'd like to list some shortcomings here:

1. The GRACE data is limited here for its coarse spatial and temporal resolution, that might mask the small-scale changes in water storage in extreme events of TCs or weather condition, and further, the incongruity between the GRACE (solution of 300 km, a pixel about 90,000 km²) and the whole assessed area of the Najd sub-basin (200,000 km², just about 2 pixels). The method might be updated to solve this problem first;

2. The contributions of TCs to GRACE change is also treated too simple because the overlaying of cyclone tracks and basin were not matched well and it reads like there was only event (Fig. 1)

The manual calibration of hydrodynamic model introduces biases and affect the predictive reliability of the model is always not so suitable;

3. The monitoring of groundwater in Najd region lacks the temporal continuity that can lead to the uncertainties to capture the immediate contribution of TCs to recharge groundwater;

4. The whole analysis process reads a bit too simple, and some other important factors should be mentioned, such as the movement of aquifers what perhaps influencing the GRACE, the contribution of rainfall events before the TCs, and so on.

I, personally, feel the manuscript does not fine enough to acceptable, and suggest the authors to improve the study and resubmit it afterwards.

Communications Earth & Environment is committed to improving transparency in authorship. As part of our efforts in this direction, we are now requesting that all authors identified as 'corresponding author' create and link their Open Researcher and Contributor Identifier (ORCID) with their account on the Manuscript Tracking System prior to acceptance. ORCID helps the scientific community achieve unambiguous attribution of all scholarly contributions. You can create and link your ORCID from the home page of the Manuscript Tracking System by clicking on 'Modify my Springer Nature account' and following the instructions in the link below. Please also inform all co-authors that they can add their ORCIDs to their accounts and that they must do so prior to acceptance.

If you experience problems in linking your ORCID, please contact the Platform Support Helpdesk.

Version 1:

Decision Letter:

Dear Professor Sultan,

Your revised manuscript titled "Intensifying tropical cyclones in the Arabian Sea replenish depleting aquifers" has now been

seen by our reviewers, whose comments appear below. In light of their advice we are delighted to say that we are happy, in principle, to publish a suitably revised version in Communications Earth & Environment.

We therefore invite you to revise your paper one last time to comply with our format requirements and to maximise the accessibility and therefore the impact of your work.

EDITORIAL REQUESTS:

****Please take care to match our formatting and policy requirements. We will check revised manuscript and return manuscripts that do not comply. Such requests will lead to delays. ****

SUBMISSION INFORMATION:

OPEN ACCESS:

Communications Earth & Environment is a fully open access journal. Articles are made freely accessible on publication. For further information about article processing charges, open access funding, and advice and support from Nature Research, please visit <https://www.nature.com/commsenv/open-access>

Link Redacted

Best regards,

Jose Luis Iriarte Machuca, PhD
Editorial Board Member
Communications Earth & Environment

Alireza Bahadori, PhD
Associate Editor
Communications Earth & Environment
Consulting Editor
Communications Sustainability

REVIEWERS' COMMENTS:

Reviewer #1 (Remarks to the Author):

The revised manuscript addresses well the comments and suggestions made from the first review.

Reviewer #3 (Remarks to the Author):

The manuscript describes the contribution of intensifying Tropical Cyclones (TCs) on the groundwater in the Arabian Peninsula (AP). A case show that the Cyclone Mekunu (2018) alone delivered 30 km³ of precipitation inland and in a net

groundwater recharge about 3.2 km³ in the Najd sub-basin. It is meaningful to understand the impacts of climate change on the resources and environment.

The clue of manuscript is clear with precipitation and GRACE considering the movement of aquifers and the relating rainfall events. All responses explained the untold parts of story and reasons of revision. This phenomenon sounds reasonable logically if we consider the process of precipitation-infiltration-groundwater.

Some shortcomings are from the data itself, but the reported "previous studies reported that the smallest resolvable basin size for GRACE is approximately 63,000 km² with an uncertainty of 2 cm" does not mean "63,000 km²" is the smallest pixel in analysis. Anyway, the results are reasonable and it would, perhaps, improve the water management in some areas prone to TCs.

Reviewer #1 Comments in (black) and our Response in (blue):

The article, “Intensifying tropical cyclones in the Arabian Sea replenish depleting aquifers”, by Hassan Saleh and co-authors is topical, given widespread concern around the consequences of the amplification of precipitation extremes (e.g. Fischer and Knutti, 2016) and ‘global-scale’ groundwater depletion observed most persistently in drylands (e.g. Jasechko et al., 2024). The manuscript describes well a thoughtful analysis showing the contributions of extreme precipitation events - tropical cyclones – to episodic groundwater replenishment in a large, arid and ungauged basin (Najd sub-basin). The authors draw upon a range of techniques and data - albeit dominated by remote sensing, to convincingly describe this (episodic) connection. It is pleasing to see the argument further rooted in ground-based observations (piezometry) presented in Figures 5 and S3.

We are grateful for Reviewer 1 encouraging comments and for recognizing the significance of our work in quantifying recharge from extreme precipitation events in arid regions. We appreciate the reviewer’s thoughtful evaluation of the methodology and findings. These comments have helped guide revisions that we believe substantially improve the manuscript. Below, we address each of Reviewer 1 remarks in detail.

Main comments – which are minor:

Authors should incorporate in a revised manuscript a more explicit discussion of:

(1) their conceptual model – based on the wide range of techniques they have employed - of how extreme rainfall is conveyed to unconfined and confined aquifers (e.g., focused versus diffuse recharge).

We did. We clarified our conceptual model of recharge in the revised manuscript (Section 4.2). We revised the text to indicate that recharge to Aquifer A primarily occurs locally through wadis. In contrast, recharge to the confined Aquifers (B, C, and D) is focused at the southern outcrops and is associated with faults, fractures, and karst features. Following recharge, groundwater likely flows through a piston-flow mechanism and discharges at Umm as Samim sabkha and dispersed springs northeast of the Najd. While diffuse recharge is very slow (0.5 m y^{-1}) (Muller, 2012), preferential groundwater flow is rapid along highly solution-channeled fractures and fault systems (Clark et al., 1987). Groundwater levels (GWL) in wells located approximately 50 km from recharge areas exhibit delayed responses of at least 6-10 months following cyclone landfall (Fig. 5d). Vertical leakage from Aquifers A and B downward into deeper Aquifers C and D is not possible in the central Najd due to higher hydraulic head pressures in these deeper confined aquifers (Müller et al., 2016). Conversely, upward leakage from Aquifer D into the overlying Aquifers and groundwater mixing along faults are possible (Al-Mashaikhi et al., 2012), further emphasizing the role of preferential flow and upward leakage as the main recharge mechanisms. **Refer to lines 249-263 in the revised manuscript.**

(2) its consequences for the uncertainty in the (volumetric) measurement of recharge. On this second point, Table 1 documents well (appropriately) uncertainty associated with the computation of recharge derived from TCP. In section 4.4, there are also thoughtful reflections on the challenges and uncertainties in estimating fluxes and storage changes in the Najd sub-basin.

We did. We revised the text (Sections 4.2 and 4.4) to clarify how uncertainty arises in volumetric recharge estimates. We now explain that GRACE-based recharge estimates integrate all infiltrated water below the evaporation depth, including water still percolating slowly through the unsaturated zone and water already reaching the water table through preferential flow. Additionally, we acknowledge uncertainties from the hydrodynamic modeling assumptions, particularly the SEC model’s uniform soil depths and the model’s

inability to simulate focused recharge through fractures or karst features. **Refer to the revised manuscript's lines 184-187 and 292-296.**

(3) The paper would be importantly improved if it included a greater synthesis of the evidence outlining (essentially) the ‘coarseness’ of the analysis - explaining why they consider groundwater recharge can be estimated (quantified) for very substantial water surpluses from TCP (e.g. Mekunu, 2018) but evidently not estimated robustly from less voluminous TCP events. Is this strictly a signal-to-noise ratio in GRACE Δ TWS? If so, how does that square with their claim – see minor comment #1 below – to provide a robust framework for quantifying recharge in ungauged arid basins worldwide?

We did. In the revised manuscript (Section 4.3), we explained that GRACE can detect total water storage (TWS) changes greater than 1-1.5 cm of equivalent water height for each native resolution grid. Cyclone Mekunu exceeded this threshold, and its impact was observed in all three GRACE solutions. In contrast, recharge from Cyclone Keila and TC 2020 likely falls within GRACE’s noise range, which explains the discrepancy between the modeled and GRACE-based estimates for these lower-magnitude events. As a result, we concluded that recharge from lower-magnitude events is more reliably captured by our hydrodynamic model, which is calibrated against surface observation, than by GRACE. We also agree with the reviewer’s observation pertaining to the overstated use of the term “robust framework” in the Abstract. The term was omitted in the revised Abstract. **Refer to lines 270-283 in the revised manuscript.**

Minor comments:

(4) In an otherwise carefully worded manuscript, the claim on lines 20-21 in the Abstract – “Our approach provides a robust framework for quantifying recharge in ungauged arid basins worldwide...” is – to my mind - overstated.

We agree. We omitted “robust” in the revised Abstract. **Refer to line 20 in the revised manuscript.**

(5) The authors have employed a range of methods (e.g. GRACE Δ TWS, piezometry, numerical modelling) to address observational constraints (i.e. ungauged catchments) in an arid basin that is of scale aligned to the use of GRACE Δ TWS without incurring the considerable uncertainty of disaggregating and downscaling GRACE Δ TWS.

We agree with Reviewer 1. We selected a large, arid watershed that receives frequent TC precipitation and is large enough to align with GRACE data resolution.

(6) The authors describe on lines 31 to 33 to “... the extremely severe cyclonic storm (Mekunu, 2018) unleashed 617 mm of torrential rain measured in the coastal city of Salalah. Mekunu caused floods that impacted the dunes field in the heart of the AP and formed lakes in the Rub Al-Khali desert (Supplementary Figure 1), which lasted for weeks.” I could find no reference to the appearance of lakes in Supplementary Figure 1.

We clarified in the revised manuscript that Supplementary Figure 1 refers to the general location of the Rub Al-Khali desert and not to the ephemeral lakes. The lakes are too small to be visible on this large-scale map. We added a citation supporting the formation and distribution of these lakes following Cyclone Mekunu. **Refer to ref 7, line 33.**

(7) The authors assert on lines 34 to 35 that, “TCs could potentially provide an opportunity to replenish depleting groundwater resources in the southern AP and other regions vulnerable to TCs including

Australia, the southwest United States, and the north African regions.” This also occurs in southern Africa – tropical cyclone Eline in February 2000 as reported by Cuthbert et al. (2019) in Namibia.

We updated the manuscript to reference TCs replenishing aquifers in sub-Saharan countries and western Africa. **Refer to lines 35-36 in the revised manuscript.**

(8) On line 200, the authors estimate (infer) “groundwater flow rates”. I suggest that the authors more precisely refer to these as “average linear velocities” to differentiate them from Darcy fluxes.

We have changed the context of this part to refer to calculating groundwater flow rates following recharge and the increase in gradient instead of our previous inference of average linear velocities. **Refer to line 233 in the revised manuscript.**

Editorial suggestions:

Line 107: The authors refer to “UER” aquifers without explicitly defining what the acronym “UERC” is. We did this in the revised text. We defined “UER” and referred to its spatial distribution and cross-section in Supplementary Figure 2, where Aquifers B, C, and D constitute Umm Er Radhuma (UER) formation. **Refer to lines 111-112 in the revised manuscript.**

Line 125: Similarly, the authors refer to “CNs” without defining this acronym.

We did this in the revised text. We defined “CNs” as the SCS curve number method for estimating rainfall partitioning. **Refer to lines 130-131.**

Line 135: Similarly, the authors refer to “SEC” model without explicitly defining what the acronym “SEC” is. This occurs later on line 351.

We did this in the revised text. **Refer to lines 51-52 in the revised manuscript.** Also, refer to **Supplementary Note 3**, which provides the equations, parameters, and data used for simulating soil evaporation.

Line 177: Replace “were” with “are”?

We did. **Refer to Line 188 in the revised manuscript.**

Line 200: Replace “meters” with “m”?

We did. **Refer to line 243 in the revised manuscript.**

Line 310: The authors refer to the hydrodynamic model, RiverFlow2D, without providing a reference or source (e.g. hydronia.com?) for this model.

We added a reference to the first mention of RiverFlow2D. **Refer to line 58 in the revised manuscript.**

Cited references:

Cuthbert, M.O., R.G. Taylor, G. Favreau, M.C. Todd, M. Shamsudduha, K.G. Villholth, A.M. MacDonald, B.R. Scanlon, D.O.V. Kotchoni, J.-M. Vouillamoz, F.M. A. Lawson, P.A. Adjomayi, J. Kashaigili, D. Seddon, J.P.R. Sorensen, G.Y. Ebrahim, M. Owor, P.M. Nyenje, Y. Nazoumou, I. Goni, B.I. Ousmane, T. Sibanda, M.J. Ascott, D.M.J. Macdonald, W. Agyekum, Y. Koussoubé, H. Wanke, H. Kim, Y. Wada, M.-H. Lo, T. Oki, N. Kukuric, 2019. Observed controls on resilience of groundwater to climate variability in sub-Saharan Africa. *Nature*, Vol. 572, 230–234.

Fischer, E.M. and Knutti, R. (2016) Observed heavy precipitation increase confirms theory and early models. *Nature Climate Change* Vol. 6, 986–991.

Jasechko, S., Seybold, H., Perrone, D., Fan, Y., Shamsudduha, M., Taylor, R.G., Fallatah, O. and Kirchner, J.W. (2024) Rapid groundwater decline and some cases of recovery in aquifers globally. *Nature*, Vol. 625, 715-721.

Reviewer #2 Comments in (black) and our Response in (blue):

This paper is a great contribution. It has been known that these cyclones are probably contributing to recharge in these arid regions, but it's been difficult to quantify. Using GRACE data and the hydrology modeling were good ways to try to get at these numbers. I have some comments below, especially about interpreting how the recharge is affecting the confined aquifers (from which most of the pumping is occurring). I think this paper should be accepted for publication, but my comments below should be addressed.

We thank Reviewer 2 for his/her supportive comments and constructive suggestions. We carefully considered all his/her comments, revised the manuscript accordingly, and provided detailed responses below.

(1) Line 33. As written, it made me look to the figure for where this lake was in the Rub Al Khali, only to be disappointed not to find it.

We clarified in the revised manuscript that Supplementary Figure 1 refers to the general location of the Rub Al-Khali desert and not to the ephemeral lakes. The lakes are too small to be visible on this large-scale map. We added a citation supporting the formation and distribution of these lakes following Cyclone Mekunu. **Refer to ref 7, line 33.**

(2) Figure 5 is not clear which wells are unconfined vs have Tritium detected, as the symbol is nearly identical.

We updated Fig. 5 to distinguish between wells in the unconfined aquifer vs wells where Tritium was detected.

(3) Line 197. Although Tritium may have been detected, change in head in confined aquifers are due to a pressure wave, not the displacement of water. I can't imagine all the water in the aquifers being displaced by the rain from one cyclone. It must have been many events since the 1960s. So Tritium is not evidence that this one cyclone recharged the confined aquifer outcrops substantially.

We agree that rapid groundwater level (GWL) responses in confined aquifers are typically attributed to pressure wave propagation, but this might not be the case in our study area. We cited additional evidence in the revised text. We applied the pressure front equation (Supplementary Note 2), and estimated an average transient time of 24.3 days for a pressure front to reach the monitoring wells following recharge. Thus, if a pressure front propagated, it would have happened 9 months before the first recorded rise in GWL in central Najd. Moreover, a pressure front occurs instantaneously and decays shortly, and thus cannot explain the observed continuous rise over 4.5 years. Following vertical recharge, the head gradient must have increased, leading to a rise in pressure and pushing groundwater in a piston-flow mechanism toward pumping-induced cones of depression in the central Najd. A similar process was described for the Lez karst system, where deep groundwater is pushed by newly infiltrated water, following intense rainfall, towards the Lez spring (Bicalho et al., 2019).” **Refer to lines 221-231 in the revised manuscript.**

We also clarified in the revised text that we do not use Tritium as evidence of recharge from Mekunu, but rather as evidence of modern recharge to the confined aquifers. Tritium was detected in water samples collected during field campaigns in 2008 in the Najd (Fig. 5b) (Al-Mashaikhi et al., 2012). **Refer to lines 213-217 in the revised manuscript.**

(4) Lines 200 through 205. Such high flow rates in karst aquifers are for unconfined aquifers with spring outlets. The UER aquifers are confined with no outlets except pumping wells today. Flow rates would be

orders of magnitude smaller. I worked with Thomas Muller on his PhD thesis (you cite him in SI) in this area. His groundwater model would have produced velocity estimates that could be used here. These were used to estimate travel times compared with his environmental tracer data. As a rough estimate the groundwaters move 500 km over 50k years. This translates to 10 meters per year.

We revised the manuscript to acknowledge two types of flow in the UER aquifers: diffuse flow through the porous media with low flow rates (Muller, 2012) and a more rapid flow through preferred pathways (faults and Karst). The presence of dual (slow diffuse and turbulent) flow regimes was reported from several karst (Caetano Bicalho et al., 2012; Driscoll et al., 2002; Worthington and Foley, 2021) and non-karst (Abdelmohsen et al., 2019) groundwater systems. These conditions are prominent in the unsaturated zone of the UER aquifers and consistent with reported transmissivity values exceeding $10,000 \text{ m}^2 \text{ day}^{-1}$ measured in Najd. Comparable rapid responses have been documented in other confined karst aquifers, such as the Edwards Aquifer (Texas, USA) (Kuniansky and Ardis, 1997) and the Madison Aquifer (South Dakota, USA) (Driscoll et al., 2002). Following vertical recharge and rapid percolation through preferential flow paths, the head gradient must have increased, pushing groundwater in a piston-flow mechanism toward pumping-induced cones of depression in the central Najd and springs further northeast (Sultan et al., 2008). A similar process was described for the Lez karst system, where deep groundwater is pushed by newly infiltrated water following intense rainfall towards the Lez spring (Bicalho et al., 2019). In the Najd, groundwater discharges into the Umm as Samim sabkha (Fookes and Lee, 2009; Muller, 2012). Several springs were mapped downstream from the recharge area in the Rub Al-Khali desert and were found to have stable isotope signatures matching deep groundwater from artesian wells in the confined aquifers (Sultan et al., 2008). **Refer to lines 221-238 in the revised manuscript.**

(5) It may be that recent pumping is pulling the modern-water front in UER inland faster, or that there is some leakage from above that contributes enough Tritium to be detected. It would be helpful to show some pumping data over time to demonstrate that the head changes in the confined aquifers are not due to changes in pumping rates. For example, did the heavy rains lead to less pumping for some time?

We have clarified in the revised manuscript that we have contacted the Ministry of Water Resources in Oman, who confirmed that there was no change in pumping activity before or after Cyclone Mekunu's landfall, nor did we observe a decrease in agricultural activity in the area following landfall (Supplementary Figure 4). Therefore, the substantial rise in groundwater levels is not attributed to a temporary reduction in pumping. However, we agree with Reviewer 2 that persistent pumping has created cones of depression in the central Najd, some exceeding 50 meters in depth (Müller et al., 2016), which increased the hydraulic gradient and thus contributed to a higher flow rate, especially through preferential flow paths from recharge zones. **Refer to lines 207-212 in the revised manuscript.**

(6) It must be true that the mass increase has occurred nearly all at the water table and above in the unsaturated zone. So, using specific yield values for the unconfined aquifers (and storativity values for the confined aquifers) in conjunction with water level changes might also produce some crude estimate of change in storage for comparisons. If you can estimate the transmissivity and storativity of the UER aquifers, could you not calculate the propagation wave of an instant recharge event at the outcrop as it extends toward and reaches the wells inland?

We agree that using specific yield and storativity values with water level changes offers a useful independent estimate of groundwater storage changes. However, several factors complicate these estimations; 1) large variations in the reported values of specific yield (0.4-10%) (Al-Mashaikhi, 2011), and transmissivity (tens to $>10,000 \text{ m}^2 \text{ day}^{-1}$) (Müller et al., 2016), 2) assigning most monitoring wells as Tertiary (consisting of confined and unconfined aquifers) or UER instead of specific aquifers, 3) a small

number of wells that cover only the central part of the Najd, and 4) recharge to Aquifer A is monitored only by two wells at the same location with 5 years gap in measurements. Therefore, we were unable to calculate meaningful changes in storage. In the revised text, we recommend future field studies to obtain site-specific aquifer parameters, enable the validation of storage estimates, and provide a summary of these limitations and recommendations. **Refer to revised Supplementary Note 2, lines 71-78.**

We have calculated the transient time for a pressure front to propagate in the confined aquifers from the recharge areas to the central Najd, where heavy pumping occurs. The pressure front equation (Chesnaux, 2018) was applied using the average transmissivity T and storativity S of aquifers B and C, with a travel distance r of 50 km. The average transient time was 24.3 days for a pressure front to reach the monitoring wells following recharge. **Refer to lines 221-231 in the revised manuscript and lines 79-82 in the revised Supplementary Note 2.**

(7) It seems that the areal extent of the GRACE anomaly is mostly over the A and B aquifers rather than the outcrops of the C and D aquifers. So in spite of the peak of the anomaly being over the C and D outcrops, the majority of the mass and recharge must have been to the B and especially the A aquifer. Which leads to the question--will increasing heads in the surficial aquifer propagated downward to the unconfined aquifers over time, or are they too hydraulically disconnected to make a difference?

We clarified this point in the revised manuscript (section 4.2). We agree that recharge to aquifer A would be widespread in the Najd, mostly where wadis cut through the aquifer and where it is unconfined. However, based on observed hydraulic head relationships, downward leakage from the surficial aquifers (A and B) to the deeper confined aquifers (C and D) is not hydraulically possible in the central Najd (Müller et al., 2016). In these areas, the pressure heads in Aquifers C and D are generally higher than in the overlying aquifers. On the contrary, groundwater is thought to leak upward from the deeper to the overlying aquifers along faults, as Al-Mashaikhi et al. (2012) suggested. Furthermore, the GRACE anomaly peak over the UER outcrops supports the interpretation that recharge to the confined aquifers occurs in the southern outcrop zones, increases the pressure head, and leads to the displacement of groundwater to the lower pressure areas in the central Najd. **Refer to lines 210-212 and lines 249-263 in the revised manuscript.**

Reviewer #3 Comments in (black) and our Response in (blue):

The impacts of climate change on evolution of land surface is always a widely- seeking topic nowadays. Most results show the negative results induced by climate change. This manuscript describes the intensification of Tropical cyclones (TCs) on the water resources in arid area, especially for the aquifers, and the results of more recharge of groundwater is quite clear and helpful for the water management in arid region. However, the study has several limitations in its methodology.

We thank the reviewer for recognizing the importance of this work for water resource management in arid regions. We have carefully considered the reviewer's comments regarding the limitations of the study and have revised the manuscript accordingly to address these concerns.

(1) The GRACE data is limited here for its coarse spatial and temporal resolution, that might mask the small-scale changes in water storage in extreme events of TCs or weather condition.

We considered the concerns of Reviewer 3. Regarding the potential influence of GRACE's coarse temporal resolution on capturing non-TC precipitation events that could affect our analysis, we examined daily precipitation data and found that non-TC rainfall contributed less than 5% of the total monthly precipitation during the month of each cyclone's landfall. This confirms that the GRACE signal during these periods is dominated by TC-related precipitation. **Refer to lines 90-93 in the revised manuscript.**

(2) and further, the incongruity between the GRACE (solution of 300 km, a pixel about 90,000 km²) and the whole assessed area of the Najd sub-basin (200,000 km², just about 2 pixels). The method might be updated to solve this problem first;

We considered the concerns of Reviewer 3 in the selection of the study area and the investigated TCs. Regarding the coarse spatial resolution of GRACE, we have stated that previous studies reported that the smallest resolvable basin size for GRACE is approximately 63,000 km² with an uncertainty of 2 cm in equivalent water height, which decreases when catchment size increases (Vishwakarma et al., 2018). The Najd subbasin is more than three times larger than this threshold, making it suitable for GRACE-based groundwater recharge estimation from TCs that produce sufficient mass change. For basins of this size, the minimum resolvable TWS anomaly is 1-1.5 cm, equivalent to ~2-3 km³ (Famiglietti and Rodell, 2013; Vishwakarma et al., 2018). **Refer to lines 276-283 in the revised manuscript.**

(3) The contributions of TCs to GRACE change is also treated too simple because the overlaying of cyclone tracks and basin were not matched well and it reads like there was only event (Fig. 1)

We have clarified in section 3.1 that the tracks shown in Fig. 1 represent the centers of each cyclone, and explained in section 5.1 that landfall is considered when the TC's center intersects land; however, its impact on land can be observed even when the center is 110 km offshore. The three tropical cyclones highlighted in this study produced the highest rainfall amounts over the Najd subbasin (Fig. 2c) and delivered rainfall as far as 800 km inland (Fig. 1c, d, and e), while rainfall from other cyclones was minor. Additionally, some cyclones shown in Figure 1a occurred during the 1990s and fall outside the scope of this study, which focuses on the GRACE data coverage (2002–2021). **Refer to lines 66-67 and lines 312-314 in the revised manuscript.**

(4) The manual calibration of hydrodynamic model introduces biases and affect the predictive reliability of the model is always not so suitable

We have clarified in section 4.4 that manual calibration of the hydrodynamic model may introduce some bias and affect predictive accuracy. However, the model results demonstrate satisfactory performance, with a Nash–Sutcliffe Efficiency (NSE) of 0.69 and a PBIAS of -3.42%, both of which fall within acceptable performance thresholds (Moriassi et al., 2007). **Refer to lines 267-269 in the revised manuscript.**

(5) The monitoring of groundwater in Najd region lacks the temporal continuity that can lead to the uncertainties to capture the immediate contribution of TCs to recharge groundwater;

We agree with the reviewer regarding the limitation of temporal continuity in groundwater level data in the Najd subbasin, which is a common challenge in arid regions. Therefore, we avoided overinterpreting short-term fluctuations and instead used the available data to illustrate accumulated recharge over time in the central Najd. This is reflected in Figure 5d and Supplementary Figure 3, which presents water level data from 27 monitoring wells in the confined aquifers.

(6) The whole analysis process reads a bit too simple, and some other important factors should be mentioned, such as the movement of aquifers what perhaps influencing the GRACE,

We did, we revised section 4.2 to include a more detailed description of the conceptual model for recharge to unconfined and confined aquifers in the Najd and the hydrogeologic controls on groundwater flow. **Refer to lines 249-263 in the revised manuscript.**

(7) the contribution of rainfall events before the TCs, and so on.

We did. We have clarified in Supplementary Note 1 and Fig. 2c that rainfall from TCs dwarfs any other rainfall source during the study period. Rainfall outside of TC events is typically negligible (<5% of the total monthly precipitation) during the month of each cyclone's landfall due to the aridity of the region, and does not influence recharge. These additions help address the complexity of the hydrological setting and further justify the analytical focus on major TC events. **Refer to lines 90-93 in the revised manuscript.**

I, personally, feel the manuscript does not fine enough to acceptable, and suggest the authors to improve the study and resubmit it afterwards.

We thank the reviewer for his/her valuable and critical comments on the manuscript. We have carefully considered each point and made substantial revisions to address the identified limitations. Our responses to the specific comments are provided, and the corresponding changes have been incorporated throughout the revised manuscript.

References

- Abdelmohsen, K., Sultan, M., Ahmed, M., Save, H., Elkaliouby, B., Emil, M., Yan, E., Abotalib, A.Z., Krishnamurthy, R.V., Abdelmalik, K., 2019. Response of deep aquifers to climate variability. *Science of The Total Environment* 677, 530–544. <https://doi.org/10.1016/j.scitotenv.2019.04.316>
- Al-Mashaikhi, K., 2011. Evaluation of groundwater recharge in Najd aquifers using hydraulics, hydrochemical and isotope evidences. Friedrich-Schiller-Universität Jena.
- Al-Mashaikhi, K., Oswald, S., Attinger, S., Büchel, G., Knöller, K., Strauch, G., 2012. Evaluation of groundwater dynamics and quality in the Najd aquifers located in the Sultanate of Oman. *Environmental Earth Sciences* 66, 1195–1211. <https://doi.org/10.1007/s12665-011-1331-2>
- Bicalho, C.C., Batiot-Guilhe, C., Taupin, J.D., Patris, N., Van Exter, S., Jourde, H., 2019. A conceptual model for groundwater circulation using isotopes and geochemical tracers coupled with hydrodynamics: A case study of the Lez karst system, France. *Chemical Geology* 528, 118442. <https://doi.org/10.1016/j.chemgeo.2017.08.014>
- Caetano Bicalho, C., Batiot-Guilhe, C., Seidel, J.L., Van Exter, S., Jourde, H., 2012. Geochemical evidence of water source characterization and hydrodynamic responses in a karst aquifer. *Journal of Hydrology* 450–451, 206–218. <https://doi.org/10.1016/j.jhydrol.2012.04.059>
- Chesnaux, R., 2018. Avoiding confusion between pressure front pulse displacement and groundwater displacement: I llustration with the pumping test in a confined aquifer. *Hydrological Processes* 32, 3689–3694. <https://doi.org/10.1002/hyp.13279>
- Clark, I.D., Fritz, P., Quinn, O.P., Rippon, P.W., Nash, H., Al Said, S., 1987. Modern and fossil groundwater in an arid environment: A look at the hydrogeology of Southern Oman.
- Driscoll, D.G., Carter, J.M., Williamson, J.E., Putnam, L.D., 2002. Hydrology of the Black Hills Area, South Dakota (No. 2). US Department of the Interior, US Geological Survey.
- Famiglietti, J.S., Rodell, M., 2013. Water in the Balance. *Science* 340, 1300–1301. <https://doi.org/10.1126/science.1236460>
- Fookes, P.G., Lee, E.M., 2009. Desert environments of inland Oman. *GeologyToday* 25, 226–231. <https://doi.org/10.1111/j.1365-2451.2009.00735.x>
- Kuniansky, E.L., Ardis, A.F., 1997. Hydrogeology and ground-water flow in the Edwards-Trinity aquifer-system, west-central, Texas, Professional Paper. U.S. Geological Survey.
- Moriasi, D.N., Arnold, J.G., Liew, M.W.V., Bingner, R.L., Harmel, R.D., Veith, T.L., 2007. Model Evaluation Guidelines for Systematic Quantification of Accuracy in Watershed Simulations. *Transactions of the ASABE* 50, 885–900. <https://doi.org/10.13031/2013.23153>
- Muller, T., 2012. Recharge and residence times in an arid area aquifer (Doctoral thesis). Technische Universität Dresden, Dresden.
- Müller, Th., Osenbrück, K., Strauch, G., Pavetich, S., Al-Mashaikhi, K.-S., Herb, C., Merchel, S., Rugel, G., Aeschbach, W., Sanford, W., 2016. Use of multiple age tracers to estimate groundwater residence times and long-term recharge rates in arid southern Oman. *Applied Geochemistry* 74, 67–83. <https://doi.org/10.1016/j.apgeochem.2016.08.012>
- Sultan, M., Sturchio, N., Al Sefry, S., Milewski, A., Becker, R., Nasr, I., Sagintayev, Z., 2008. Geochemical, isotopic, and remote sensing constraints on the origin and evolution of the Rub Al

Khali aquifer system, Arabian Peninsula. *Journal of Hydrology* 356, 70–83.
<https://doi.org/10.1016/j.jhydrol.2008.04.001>

Vishwakarma, B.D., Devaraju, B., Sneeuw, N., 2018. What Is the Spatial Resolution of grace Satellite Products for Hydrology? *Remote Sensing* 10, 852. <https://doi.org/10.3390/rs10060852>

Worthington, S.R.H., Foley, A.E., 2021. Deriving celerity from monitoring data in carbonate aquifers. *Journal of Hydrology* 598, 126451. <https://doi.org/10.1016/j.jhydrol.2021.126451>